# Neuronal IFN-beta–induced PI3K/Akt-FoxA1 signalling is essential for generation of FoxA1$^+$T$_{reg}$ cells

Yawei Liu[1], Andrea Marin[1], Patrick Ejlerskov[1], Louise Munk Rasmussen[1], Marco Prinz[2] & Shohreh Issazadeh-Navikas[1]

Neurons reprogramme encephalitogenic T cells (T$_{enc}$) to regulatory T cells (T$_{regs}$), either FoxP3$^+$T$_{regs}$ or FoxA1$^+$T$_{regs}$. We reported previously that neuronal ability to generate FoxA1$^+$T$_{regs}$ was central to preventing neuroinflammation in experimental autoimmune encephalomyelitis (EAE). Mice lacking interferon (IFN)-β were defective in generating FoxA1$^+$T$_{regs}$ in the brain. Here we show that lack of neuronal IFNβ signalling is associated with the absence of programme death ligand-1 (PDL1), which prevents their ability to reprogramme T$_{enc}$ cells to FoxA1$^+$T$_{regs}$. Passive transfer-EAE via IFNβ-competent T$_{enc}$ cells to mice lacking IFNβ and active induced-EAE in mice lacking its receptor, IFNAR, in the brain (Nes$^{Cre}$:Ifnar$^{fl/fl}$) result in defective FoxA1$^+$T$_{regs}$ generation and aggravated neuroinflammation. IFNβ activates neuronal PI3K/Akt signalling and Akt binds to transcription factor FoxA1 that translocates to the nucleus and induces PDL1. Conversely, inhibition of PI3K/Akt, FoxA1 and PDL1 blocked neuronal ability to generate FoxA1$^+$T$_{regs}$. We characterize molecular factors central for neuronal ability to reprogramme pathogenic T cells to FoxA1$^+$T$_{regs}$ preventing neuroinflammation.

[1] Neuroinflammation Unit, Biotech Research and Innovation Centre (BRIC), Faculty of Health and Medical Sciences, University of Copenhagen, Copenhagen Biocentre, Ole Maaløes Vej 5, DK-2200 Copenhagen N, Denmark. [2] Institute for Neuropathology and Centre for Biological Signaling Studies, University of Freiburg, 79106 Freiburg, Germany. Correspondence and requests for materials should be addressed to S.I.-N. (email: shohreh.issazadeh@bric.ku.dk).

Regulatory T cells (T$_{regs}$) are essential in controlling inflammation. FoxP3$^+$T$_{reg}$ cells have been shown to prevent overactivation of the immune system in systemic and central nervous system (CNS) inflammatory diseases[1,2]. We have recently identified a new T$_{reg}$ cell population, FoxA1$^+$T$_{reg}$ cells, which played a pivotal role in regulation of CNS inflammation[3].

The outcome of CNS inflammation depends on how immune cells, vascular cells and neurons interact[4]. Neurons have evolved several mechanisms to directly interact with T cells, to protect against development of chronic CNS inflammation. Neurons have been shown to promote T-cell apoptosis through Fas/FasL interactions[5] and to produce anti-inflammatory cytokines, such as transforming growth factor-β, which in turn prevents experimental autoimmune encephalomyelitis (EAE)[6,7]. The immunoregulatory activities of neurons and their central role in controlling CNS diseases, partly by regulating CNS inflammation, have recently received profound support[3,7–9]. We have previously shown that neurons directly interact with pathogenic T cells; this subsequently led to the generation of FoxP3$^+$T$_{regs}$ with capacity to ameliorate EAE[7]. Moreover, we reported that neurons were capable of converting EAE-inducing encephalitogenic T cells (T$_{enc}$) to anti-inflammatory FoxA1$^+$T$_{regs}$ (ref. 3). Of note, mice genetically lacking interferon (IFN)-β signalling developed severe relapsing-remitting and demyelinating EAE, which subsequently was inhibited upon passive transfer of neuron-induced FoxA1$^+$T$_{regs}$ cells[3]. In support, beneficial IFNβ-treatment response in patients with multiple sclerosis (MS) was associated with increased generation of circulating FoxA1$^+$T$_{reg}$ cells[3].

Here we investigate whether neuronal IFNβ is essential for their capacity to reprogramme T$_{enc}$ cells to become FoxA1$^+$T$_{regs}$ and furthermore study the molecular signalling by which IFNβ coordinates this central immunoregulatory function of neurons. We report that IFNβ-mediated FoxA1 activation results in expression of neuronal PDL1. IFNβ activates neuronal PI3K/Akt signalling. Consequently, Akt binds to transcription factor FoxA1 that translocates to the nucleus and induces PDL1. This is indispensable for neuronal ability to reprogramme T$_{enc}$ cells and generate FoxA1$^+$T$_{regs}$, which in turn are important to combat neuroinflammation.

## Results

**Lack of FoxA1$^+$T$_{regs}$ leads to neuroinflammation in Ifnb$^{-/-}$ mice.** Previously with active EAE, we identified FoxA1$^+$T$_{regs}$ from CNS-infiltrating T cells in IFNβ-competent wild-type (WT) (Ifnb$^{+/+}$) mice. In the contrary, FoxA1$^+$T$_{regs}$ were not found in EAE mice genetically lacking IFNβ (Ifnb$^{-/-}$)[3]. We have also reported that IFNβ signalling in T cells was central for their FoxA1$^+$T$_{reg}$ cell fate determination. To exclude the role of intrinsic IFNβ signalling in T cells, we utilized IFNβ-competent T$_{enc}$ cells (MBP$_{89-101}$ reactive T$_{enc}$ cells) to transfer adoptive EAE in Ifnb$^{+/+}$ or Ifnb$^{-/-}$ mice. As previously reported for the active EAE[3], inducing adoptive transfer of EAE led to higher clinical disease score in Ifnb$^{-/-}$ mice (Fig. 1a). This was associated with defective FoxA1$^+$T$_{reg}$ (TCR$^+$FoxA1$^+$PDL1$^+$) cell generation in the CNS-infiltrating T cells in Ifnb$^{-/-}$ spinal cord in contrast to Ifnb$^{+/+}$ mice, day 30 post adoptive EAE (Fig. 1b); this indicates that despite utilizing the same WT-MBP$_{89-101}$ reactive T$_{enc}$ cells in both groups, the determining factor is the host endogenous IFNβ. Interestingly, although Ifnb$^{-/-}$ mice developed significantly higher neuroinflammation apparent by elevated total number of infiltrating T cells in the spinal cord even during remission (Fig. 1c), they had significantly lower FoxA1$^+$T$_{regs}$ compared with Ifnb$^{+/+}$ mice (Fig. 1d).

Of note, FoxA1$^+$T$_{regs}$ were often detected not only in perivascular space (Fig. 1b) but also in parenchyma adjacent to neuronal soma (Fig. 1e); this provides neurons with possibility to interact with T cells and induce their FoxA1 expression both via molecules expressed on axons or on neuronal cell bodies, in the areas that T cells usually invade the CNS remote from neuronal bodies, as well as closed to the neuronal bodies upon migrating in the parenchyma, respectively. Moreover, a striking difference was observed in neuronal PDL1 expression in the cerebellum and spinal cord, which was lacking in Ifnb$^{-/-}$ mice (Fig. 1f,g). Of interest, although PDL1 was not detectable in Ifnb$^{-/-}$ mice, FoxA1 was expressed (Fig. 1g). Of interest, the significant increase of FoxA1$^+$T$_{regs}$ in Ifnb$^{+/+}$ mice (Fig. 1b,d) was associated with neuronal co-expression of FoxA1 and PDL1, whereas lack of IFNβ was associated with loss of neuronal PDL1 expression (Fig. 1g,h) and concomitant loss of FoxA1$^+$T$_{reg}$ cells in the CNS of Ifnb$^{-/-}$ mice. These results suggested an important role for IFNβ signalling in the CNS to regulate the generation of FoxA1$^+$T$_{reg}$ cells.

**Nes$^{cre}$:Ifnar$^{fl/fl}$ mice lose ability to generate FoxA1$^+$T$_{regs}$.** To address the role of neuronal IFNβ-IFNAR signalling in regulation of CNS inflammation associated with FoxA1$^+$T$_{reg}$ cell generation, Nes$^{cre}$:Ifnar$^{fl/fl}$ mice were actively immunized with MOG$_{35-55}$ (ref. 10). Quantification of inflammatory cells infiltrating in the spinal cord of mice 35 days post immunization revealed that Nes$^{cre}$:Ifnar$^{fl/fl}$ mice developed profound neuroinflammation compared with their WT corresponding, Ifnar$^{fl/fl}$ mice (Fig. 2a,b). Similar to mice lacking genomic IFNβ, loss of brain IFNAR (IFNα/β receptor) signalling in Nes$^{cre}$:Ifnar$^{fl/fl}$ mice resulted in the lack of FoxA1$^+$T$_{reg}$-cell generation associated with elevated neuroinflammation (Fig. 2c–e). Of note, loss of neuronal IFNAR signalling led to the loss of PDL1 expression, while FoxA1 was still expressed by neurons (Fig. 2f,g). Taken together, these results strongly indicated that active neuronal IFNβ-IFNAR signalling is central for converting T$_{enc}$ cells to FoxA1$^+$T$_{reg}$ cells and hence for controlling neuroinflammation in the CNS.

**Neuronal IFNβ signalling is essential to generate FoxA1$^+$T$_{regs}$.** Although neurons were found to be able to generate FoxA1$^+$T$_{regs}$ (ref. 3) and the results above supported an active role for neuronal IFNβ signalling, the molecular mechanisms by which neurons exert such a fundamental immunoregulatory property were not known. Here we investigated whether neuronal IFNβ was involved in their T-cell-reprograming capacity. To exclude other CNS-resident cell contribution, we established primary neuronal cultures with high purity (Fig. 3a) (that is, mean ± s.d. of 98.3 ± 0.28%, $n = 3$) and compared them with astrocytes and microglial cells in regard to their capacity to generate FoxA1$^+$T$_{reg}$ cells upon interacting with T$_{enc}$ cells. As expected, cerebellar granular neurons (CGNs) changed phenotype of activated T$_{enc}$ cells and generated FoxA1$^+$T$_{regs}$, whereas other CNS innate antigen-presenting cells (APCs); microglia and astrocytes did not exert these properties (Fig. 3b). We have previously shown that neurons remain electrically active in culture[7]. To investigate whether neuronal electrical activity is required for the induction of FoxA1$^+$T$_{regs}$, neurons were silenced by tetrodotoxin (TTX). TTX is a potent toxin that specifically binds to voltage-gated sodium channels and blocks the flow of sodium ions through the channel, thereby preventing action potential generation and propagation[11], and silences neurons[12]. TTX-silenced neurons lost the capacity to induce generation of FoxA1$^+$T$_{regs}$ (Fig. 3c).

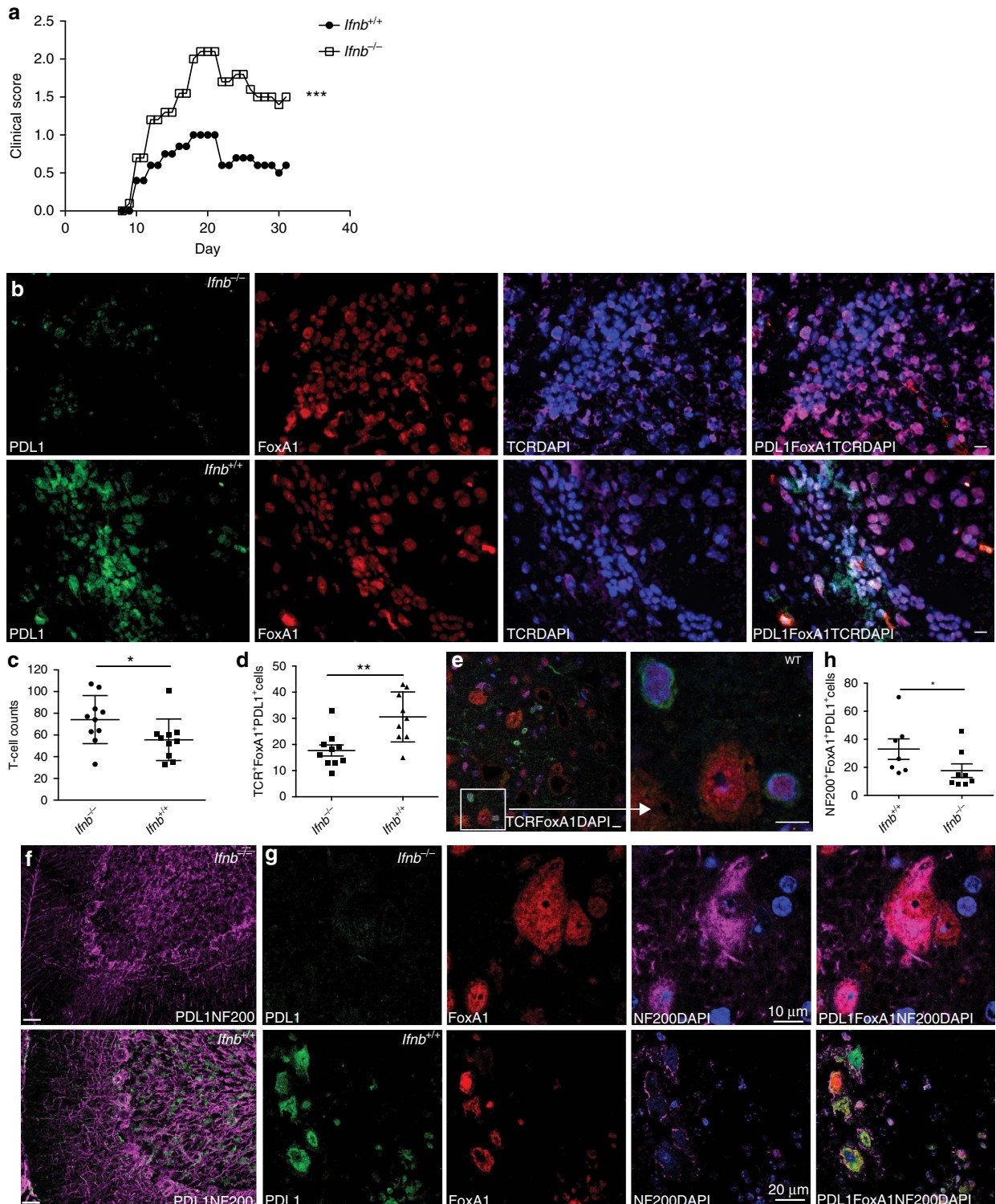

**Figure 1 | Adoptive transfer of $T_{enc}$ cells to $Ifnb^{-/-}$ mice causes elevated neuroinflammation associated with defective FoxA1[+] $T_{reg}$ cell generation.**
(**a**) Adoptive transfer of $MBP_{89-101}$ $T_{enc}$ cells to C57BL/10.RIII mice, EAE score from $Ifnb^{+/+}$ and $Ifnb^{-/-}$ mice, $n = 24$ mice per group. Non-parametric Mann–Whitney test was used, ***$P < 0.001$. (**b**) Representative IF images in spinal cords from mice day 30 adoptive EAE. PDL1(green), FoxA1(red), TCR(purple) and DAPI(blue). Scale bars, 20 µm. (**c**) Quantification of number of spinal cord infiltrating T cells, (**d**) quantification of number of FoxA1[+] $T_{reg}$ cells. Graphs are mean ± s.e.m., $n = 10$ sections per group. Non-parametric Mann–Whitney test was used, *$P < 0.05$ and **$P < 0.01$. (**e**) IF images of T cells in close vicinity of neurons in spinal cords of a WT mouse, day 30 post-adaptive EAE. FoxA1 (red), TCR (purple) and DAPI (blue). Scale bars, 10 µm. (**f**) Representative IF images of neurons in cerebellums. PDL1 (green) and NF200 (purple) were stained and shown. Scale bars, 30 µm. (**g**) Closed up representative IF images of neurons in spinal cords. PDL1 (green), FoxA1 (red), TCR (purple) and DAPI (blue). (**h**) Quantification of number of NF200[+]FoxA1[+]PDL11[+] neurons in spinal cords. Graphs are mean ± s.e.m., $n = 7$-8 sections per group. Non-parametric Mann–Whitney test was used, *$P < 0.05$.

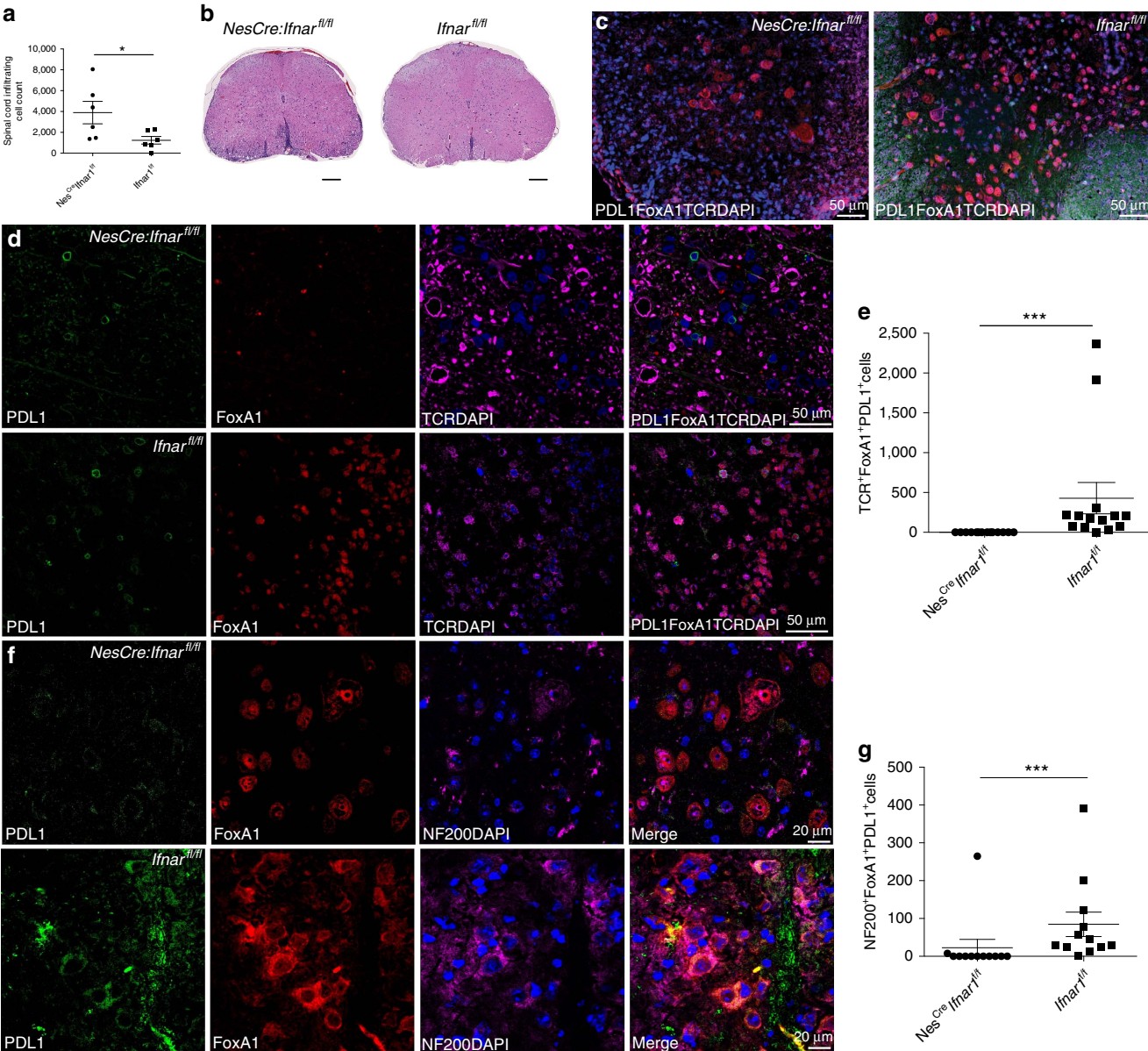

**Figure 2 | Defective neuronal IFNβ-IFNAR signalling in *Nes^cre:Ifnar^fl/fl* mice leads to loss of capacity to generate FoxA1^+ T_regs.** (**a**) Quantification of number of infiltrating inflammatory cells in spinal cords in *Ifnar^fl/fl* (WT) and *Nes^Cre:Ifnar^fl/fl* mice with active EAE. Graphs are mean ± s.e.m., n = 6 per group. Non-parametric Mann–Whitney test was used, *P < 0.05. (**b**) Representative spinal cord sections stained with H&E day 35 post EAE immunization. Scale bars, 200 μm. (**c**) Overview of IF images of spinal cords. PDL1 (green), FoxA1 (red), TCR (purple) and DAPI (blue). Scale bar, 50 μm. (**d**) Closed up IF images of representative spinal cords. PDL1 (green), FoxA1 (red), TCR (purple) and DAPI (blue). Scale bars, 50 μm. (**e**) Quantification of FoxA1^+ T_reg cells in spinal cords. Graphs are mean ± s.e.m., n = 12 sections per group. (**f**) Closed up IF images of representative spinal cords. PDL1 (green), FoxA1 (red), NF200 (purple) and DAPI (blue). Scale bars, 20 μm. (**g**) Quantification of NF200^+ FoxA1^+ PDL1^+ neurons in spinal cords. Graphs are mean ± s.e.m., n = 12 sections per group. Non-parametric Mann–Whitney test was used, ***P < 0.001.

We next examined the suppressive capacity of these neuronal-induced FoxA1^+ T_reg (nFoxA1^+ T_reg) cells *in vivo* and *in vitro*. As previously reported[3], purified nFoxA1^+ T_regs could induce significant cell death of activated T_enc cells *in vitro* (Fig. 3d). To confirm their suppressive activities *in vivo*, purified nFoxA1^+ T_regs were adoptively transferred to ears in a murine delayed-type hypersensitivity (DTH) model of tissue inflammation. Ears receiving nFoxA1^+ T_regs had significantly less inflammation (Fig. 3e). These results supported suppressive and anti-inflammatory properties of FoxA1^+ T_regs *in vivo* and *in vitro*.

We finally examined the need for endogenous neuronal IFNβ in generation of FoxA1^+ T_regs. We observed that both CGNs and

cortical neurons (CNs) from *Ifnb^+/+* neurons were able to generate FoxA1^+ T_regs, whereas neurons lacking IFNβ (*Ifnb^−/−*) lost this capacity (Fig. 3f–h). Treatment of *Ifnb^−/−* neurons with recombinant (r)IFNβ to reconstitute their defect, before co-culture with activated T_enc cells, restored their ability to generate FoxA1^+ T_regs (Fig. 3i). These data indicated that neuronal ability to convert pathogenic T_enc cells to FoxA1^+ T_reg cells depends on their endogenous IFNβ signalling. IFNβ share many functional similarities with IFNα, as they share the same receptor, IFNAR; however, they also differ in many of their functions including their different efficiencies as disease treatment. Although it is not well described how IFNβ might

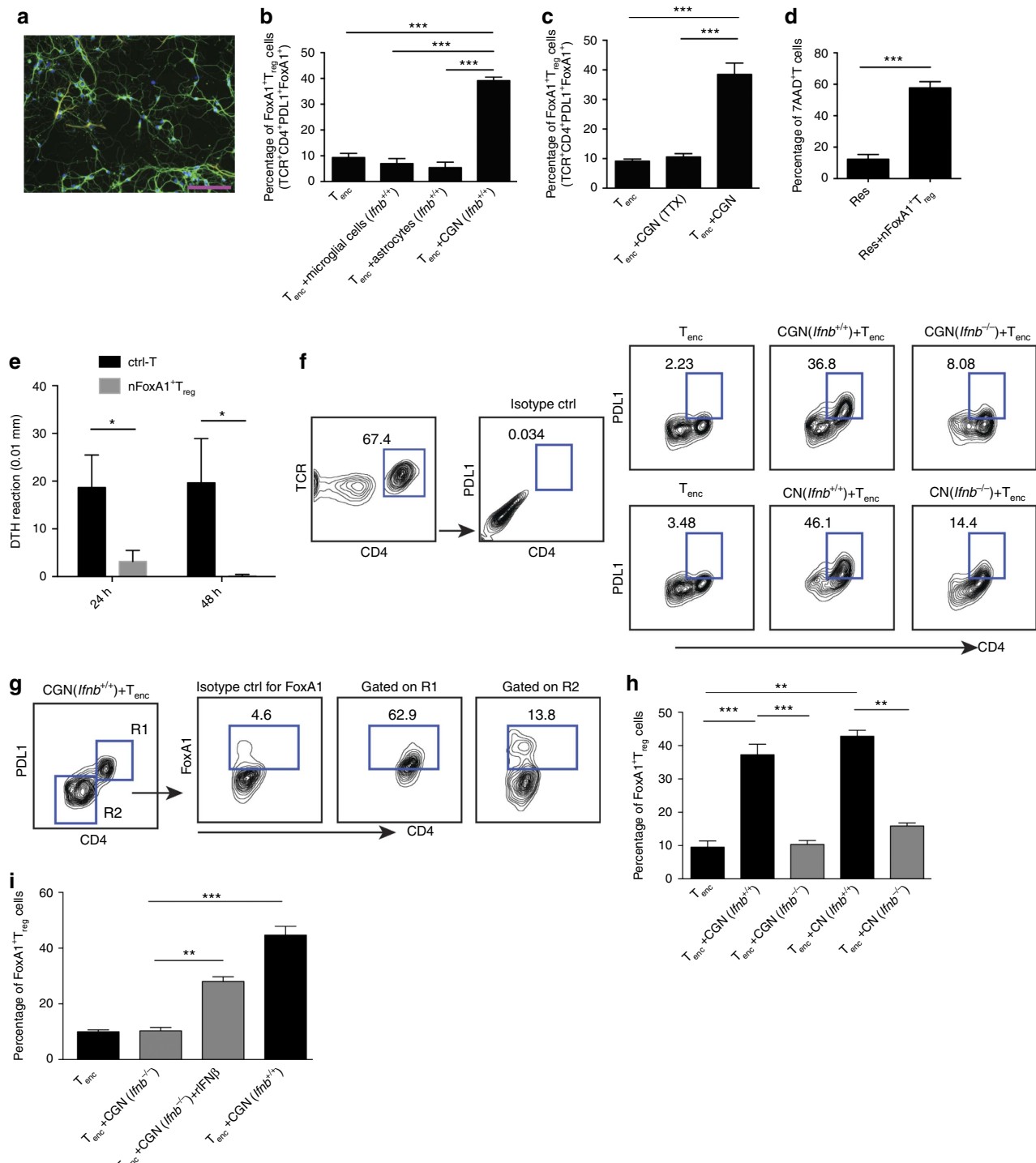

**Figure 3 | Neuronal capacity to generate FoxA1$^+$ T$_{reg}$ cells requires IFNβ.** (**a**) One representative IF image from neuronal culture was stained with anti-GFAP (yellow) and anti-β III tubulin (green). Scale bar, 100 μm. (**b**) Percentage of FoxA1$^+$ T$_{reg}$ cells were analysed by FACS after co-culture of activated T$_{enc}$ cells with CG neurons, microglial or astrocytes, respectively. Graphs are mean ± s.e.m., $n = 3$. One-way analysis of variance (ANOVA) test was used, ***$P < 0.001$. (**c**) Percentage of FoxA1$^+$ T$_{regs}$ were analysed by FACS after co-culture of T$_{enc}$ cells with CGN treated with or without TTX (1 μM) for 24 h before co-culture. Graphs are mean ± s.e.m., $n = 3$, one-way ANOVA test was used, ***$P < 0.001$. (**d**) Percentage of 7AAD$^+$ (dead) cells was analysed by FACS. Res, responder T cells were activated with plate-bound anti-CD3 (1 μg ml$^{-1}$) and soluble anti-CD28 (2 μg ml$^{-1}$) for 24 h, then co-cultured with neurons to generate FoxA1$^+$ T$_{reg}$ cells (nFoxA1$^+$ T$_{reg}$) for another 24 h. Graphs are mean ± s.e.m., $n = 3$. Unpaired Student's $t$-test was used, ***$P < 0.001$. (**e**) DTH measured as ear thickness at 24 and 48 h post injection of nFoxA1$^+$ T$_{reg}$ or control cells (ctrl-T). Graphs are mean ± s.e.m., $n = 3$ mice per group. Unpaired Student's $t$-test was used, *$P < 0.05$. (**f**) FoxA1$^+$ T$_{reg}$ cell FACS gating strategy. (**g**) Gated CD4$^{high}$PDL1$^{high}$ cells (R1) and CD4$^{low}$PDL1$^{low}$ cells (R2). (**h**) Percentage of FACS analysis of FoxA1$^+$ T$_{regs}$ after co-culture of T$_{enc}$ cells with CGN or CN. Graphs are mean ± s.e.m., $n = 3$, one-way ANOVA test was used, **$P < 0.01$ and ***$P < 0.001$. (**i**) Percentage of FoxA1$^+$ T$_{regs}$ were analysed by FACS after co-culture of T$_{enc}$ cells with CGN treated with or without rIFNβ (100 U per ml). One-way ANOVA test was used, **$P < 0.01$ and ***$P < 0.001$.

regulate IFNα, it is previously reported that IFNβ is required for production of IFNα in fibroblast[13] and we have not detected any compensatory mechanisms in neurons when only IFNβ is deleted[9]. Although IFNα might have additional or differential effects independent of IFNβ, this has not been observed related to the neuronal activity. Moreover, there are several alleles for *Ifna*, which result in the production of many active IFNα isoforms. This is imposing much more complexity to investigate the plausible role for IFNα isoforms in relation to the neuronal functions and hence not the focus of the current study.

**Generation of FoxA1$^+$T$_{regs}$ requires neuronal FoxA1 and PDL1**. Previously, we have shown that the soluble cytokine IFNβ has an anti-inflammatory role in the CNS, as evidenced by the extensive CNS inflammation, relapses and demyelination in *Ifnb$^{-/-}$* mice with EAE[14]. In addition, it was shown that treatment of T cells with exogenous rIFNβ was sufficient to induce FoxA1$^+$T$_{regs}$ (ref. 3). To understand whether soluble IFNβ produced by neurons directly affects T$_{enc}$ cells to change their phenotype to FoxA1$^+$T$_{regs}$, we utilized a transwell system to separate neurons and T cells in co-cultures, allowing free circulation of IFNβ. Separation of neurons from T$_{enc}$ cells completely diminished FoxA1$^+$T$_{reg}$ cell generation (Fig. 4a), which suggests that cell-to-cell contact is necessary for neuronal conversion of pathogenic T$_{enc}$ cells to anti-inflammatory FoxA1$^+$T$_{regs}$.

It was established that neuronal signaling molecule PDL1 inhibited cell cycle and induced apoptosis in brain tumours, namely glioblastoma[8]; however, it was not known whether it could be involved in interaction between neurons and T cells. In T cells, IFNβ treatment activates the transcription factor FoxA1 and thereby transcription of its downstream target gene *Pdl1* (ref. 3). Earlier, FoxA1 has been reported to play a role in survival of dopaminergic neurons[15]; however, it was unknown whether IFNβ has an impact on neuronal regulation of FoxA1 or whether FoxA1 targets the *Pdl1* gene in neurons. We hypothesized that upon IFNβ treatment, neurons would activate FoxA1 to induce transcription of *Pdl1*. The lack of neuronal PDL1 expression both in *Ifnb$^{-/-}$* and *Nes$^{cre}$:Ifnar$^{fl/fl}$* mice strongly supported such a scenario. We therefore examined the messenger RNA expression levels of *Foxa1* and *Pdl1* in CGNs, with or without rIFNβ treatment. *FoxA1* and *Pdl1* mRNA were significantly increased after rIFNβ stimuli (Fig. 4b,c). We also found remarkable differences in the basal levels of FoxA1 and PDL1 proteins in *Ifnb$^{+/+}$* and *Ifnb$^{-/-}$* neurons, which were further increased upon rIFNβ treatment (Fig. 4d–h). Moreover, treatment of *Ifnb$^{+/+}$* neurons with rIFNβ led to an increased translocation of FoxA1 from the cytoplasm to the nucleus (Fig. 4f,g). In addition, we identified a striking difference in the subcellular localization of FoxA1 in *Ifnb$^{+/+}$* versus *Ifnb$^{-/-}$* neurons: although FoxA1 was mainly found in the perinuclear space in *Ifnb$^{+/+}$* neurons and translocated to the nucleus upon rIFNβ treatment, in *Ifnb$^{-/-}$* neurons FoxA1 was almost exclusively nuclear (Fig. 4g). Of note, the neuronal marker NF200 was also detected in the nucleus, the precise function of which is not well defined; an explanation for this observation might be the fact that neurofilaments are used as ducking or carriage proteins to transport signalling molecules.

We next investigated whether FoxA1 and its downstream target PDL1 were necessary for neuronal capacity to generate FoxA1$^+$T$_{regs}$. PDL1 was found to be much lower in *Ifnb$^{-/-}$* neurons and rIFNβ significantly upregulated PDL1 protein in both *Ifnb$^{+/+}$* and *Ifnb$^{-/-}$* neurons as previously reported by us[8]. To study the function of neuronal FoxA1 and PDL1, knockdown (KD) of neuronal FoxA1 and PDL1 was successfully achieved using small-interfering RNAs (siRNAs) (Fig. 4i,j,l).

Importantly, neuronal FoxA1 KD led to reduced PDL1 expression (Fig. 4i,j), indicating that, as in T cells[3], *Pdl1* is a target gene for FoxA1 in neurons. Moreover, KD of FoxA1 or PDL1 in neurons before co-culture with T$_{enc}$, significantly reduced their capacity to convert T$_{enc}$ cells to FoxA1$^+$T$_{regs}$ (Fig. 4k,m). In support, blocking neuronal PDL1 signalling in the *Ifnb$^{+/+}$* neurons (using a blocking anti-PDL1 antibody) or blocking PD1 on T$_{enc}$ cells (using a blocking anti-PD1 antibody) resulted in inhibition of FoxA1$^+$T$_{reg}$ cell generation (Fig. 4n). Moreover, overexpressing PDL1 (using the pIRESII PDL1 plasmid) on *Ifnb$^{-/-}$* neurons restored their ability to generate FoxA1$^+$T$_{reg}$ cells (Fig. 4o,p). These results conclusively underscores that defective neuronal FoxA1 and PDL1 could account for the decreased or lack of ability to induce FoxA1$^+$T$_{regs}$, and that IFNβ signalling is central for activation of neuronal FoxA1 and its target gene *Pdl1*, which in turn are indispensable for the function of neurons to interact and convert pathogenic T cells to FoxA1$^+$T$_{reg}$ cells.

**IFNβ-induced PI3K/Akt regulates neuronal FoxA1 and PDL1**. It is known that JAK (Janus kinase) and a STAT (signal trans-ducer and activator of transcription) pathway is activated by IFNs; however, the importance of many other pathways in IFN-mediated signalling is also established[16–20]. Here we aimed to characterize the downstream signalling pathway for IFNβ, which regulates neuronal FoxA1-mediated transcription of *Pdl1*. We investigated different pathways activated by IFNβ, including P38 mitogen-activated protein kinases (p38 MAPK), MAPK3 (Erk1), MAPK1 (Erk2), phosphatidylinositol-3-kinases (PI3Ks) and protein kinase B (Akt). We detected significant differences regarding higher basal *Pi3k* and *Akt* mRNA and protein levels in *Ifnb$^{+/+}$* versus *Ifnb$^{-/-}$* neurons (Fig. 5a–e). Phosphorylated (p)PI3K and pAkt are signs of active signalling molecules, which were even more pronouncedly reduced in *Ifnb$^{-/-}$* neurons compared with the IFNβ-competent neurons. Reduced pPI3K/pAkt in *Ifnb$^{-/-}$* neurons were also associated with significant lower FoxA1 and PDL1 (Fig. 5b,c). Moreover, we found an abundant *Pi3k* and *Akt* transcription (Fig. 5d,e) and protein induction upon IFNβ treatment of neurons; however, here were no differences detected for pErk and pP38 (Fig. 5f). These data suggested that PI3K/Akt might act upstream of IFNβ-induced FoxA1-PDL1 regulation. To address this issue, we chemically inhibited PI3K and Akt signalling in *Ifnb$^{+/+}$* neurons, which completely eliminated IFNβ-induced FoxA1 and PDL1 expressions in neurons (Fig. 5f–h).

Next, we examined whether Akt physically binds to FoxA1 in regards to activate the pathway leading to PDL1 induction. In support of this scenario, when we overexpressed FoxA1 with a flag-tag in neuronal cell line N2A, we could immunoprecipitate (IP) FoxA1, leading to pull down of endogenous Akt upon co-IP (Fig. 5i), which was confirmed by IP of endogenous FoxA1 and co-IPing Akt upon rIFNβ treatment of N2A and primary neurons (Fig. 5j,k), respectively. Our results provided direct evidence that IFNβ-induced PI3K/Akt signalling is essential for the induction of FoxA1 and subsequently PDL1 in neurons.

**Neuronal PI3K/Akt is essential to reprogramme T cells**. We demonstrated that engagement of PI3K/Akt pathway was required for IFNβ-mediated neuronal expressions of FoxA1 and PDL1. To establish whether the PI3K/Akt signalling is required for neurons to convert T$_{enc}$ cells to FoxA1$^+$T$_{regs}$, we blocked neuronal PI3K and Akt activities using either PI3K inhibitor or KD of Akt by siRNA silencing before neurons were co-cultured with encephalitogenic T$_{enc}$ cells. Inhibition of PI3K/Akt signalling in neurons significantly reduced the

protein levels of FoxA1 and PDL1 (Fig. 5), and resulted in complete abrogation of their function to convert $T_{enc}$ cells to FoxA1$^+$T$_{regs}$ (Fig. 6a–c). Collectively, our data revealed that endogenous IFNβ signalling is central for the immunoregulatory function of neurons in which they convert pathogenic T cells to

become FoxA1$^+$T$_{regs}$. We showed that IFNβ triggers PI3K/Akt activation and Akt subsequently bound to FoxA1. This process in turn activated neuronal FoxA1 to induce PDL1. We also established that IFNβ-induced PI3K/Akt pathway and its downstream FoxA1-mediated PDL1 expression were

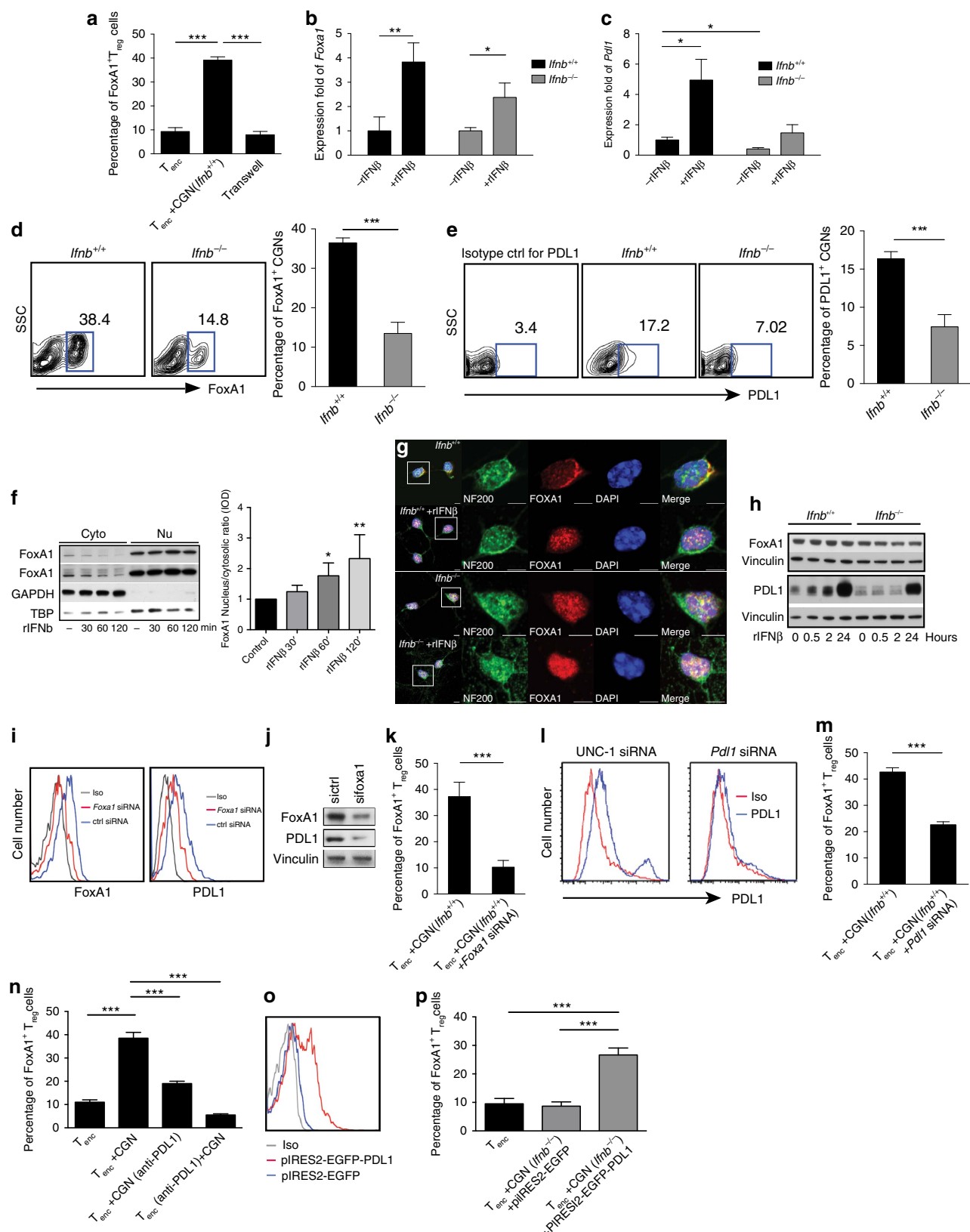

indispensable for the immunoregulatory function of neurons to generate FoxA1$^+$T$_{regs}$ (Fig. 6d).

## Discussion

The concerted action between immune cells and tissue-specific cells in their microenvironment is crucial for the outcome of inflammation. Autoimmune inflammation is a key element in numerous CNS diseases. The bidirectional interaction between autoreactive (CNS antigen-specific) immune cells with brain-resident cells could therefore be central for the fate of autoimmune-mediated inflammatory diseases such as MS. CNS-specific autoimmunity is chiefly related to how autoreactive T and B cells cause damage and/or mutually interact with CNS-resident APCs to consequently maintain chronic activation of autoreactive cells and thereby neuroinflammation[21–24]. An important role of local neurons in regulation of brain immunity is conceivable; however, it has only recently received more attention. A direct role in immune regulation was reported for neurons to promote T-cell apoptosis[5]. In support, neuronal nuclear factor, nuclear factor-κB was also shown to regulate neuroinflammation[25]. Moreover, we reported several pivotal immunosuppressive properties of neurons, in which neurons interact with autoreactive T$_{enc}$ cells and convert them to regulatory FoxP3$^+$T$_{reg}$ via production of transforming growth factor-β and expression of B7.1 (ref. 7), and/or converting them to adapt a FoxA1$^+$T$_{reg}$ fate and anti-inflammatory functions[3].

Although the accumulative evidence is reinforcing the importance of neurons in controlling CNS immunity, current understanding of the essential basic molecular mechanisms of neuronal control of neuroinflammation is sparse. Here we investigated the required molecular signalling operative in neurons, which provides them with capacity to convert pathogenic T cells to the newly identified FoxA1$^+$T$_{reg}$ cells and their impact on outcome of neuroinflammation. As mice lacking the *Ifnb* gene were deficient in generating FoxA1$^+$T$_{regs}$ in the CNS[3], we studied the function of endogenous neuronal IFNβ when challenged with IFNβ-competent T$_{enc}$ cells. Neurons defective in IFNβ production were incompetent to convert T$_{enc}$ cells to FoxA1$^+$T$_{regs}$, both *in vitro* and *in vivo*, indicating a central function of this cytokine in neuronal immunoregulatory efficacies. In support, neuronal IFNβ was shown to be critical for their anti-tumor functions to control glioblastoma[8]. Soluble IFNβ was shown to exert anti-tumour activities[16,26] and it was sufficient to convert naive T cells to FoxA1$^+$T$_{regs}$ (refs 3,27). However, blocking the direct cell-to-cell interaction between neurons and T cells prevented the conversion of T$_{enc}$ to

FoxA1$^+$T$_{regs}$, which revealed that released soluble neuronal IFNβ alone was not sufficient to convert T cells. These data suggested that endogenous neuronal IFNβ exerted paracrine and autocrine effects to provide neurons with additional signalling key molecules. In support, lack of IFNAR on neurons was leading to the similar loss of function in *Nes$^{Cre}$:Ifnar$^{fl/fl}$* mice, which strongly displayed evidence that released soluble IFNβ by neurons[9] requires active autocrine signalling, via its receptor, to synchronize neuronal immunoregulatory functions. Pleiotropic biological activities of IFNβ was reported to be mediated by triggering different signalling pathways[16]. We demonstrated that the PI3K/Akt pathway was activated downstream of neuronal IFNβ signalling. In agreement, PI3K has been shown to be responsible for several biological activities of IFNα/β in different types of cells even independent of JAK–STAT signalling[28–30]. Activation of PI3K and Akt were essential for induction of other downstream signalling molecules in neurons. We showed that Akt binds to FoxA1. As a transcription factor, the role of FoxA1 in neurodevelopmental processes and survival of dopaminergic neurons has been reported[15], but no role was established related to immunoregulatory function of post-mitotic neurons, in particular related to IFNβ signalling. Interestingly, *in vivo*, FoxA1 could be detected in neurons depleted of IFNβ or IFNAR, but lack of active signalling did prevent it from targeting the *Pdl1* gene, suggesting that activation of PI3K and Akt could modify FoxA1 activity. Indeed, insulin-like growth factor-I was reported to increase the stability of FoxA1 protein in MCF7 breast cancer cell line via Akt pathway[31].

Our data suggest that activated Akt binds to FoxA1, which might mediate its translocation to the neuronal nucleus and upon binding to the *Pdl1* promoter induces its production. In support, we have previously shown that translocation of FoxA1 to nucleus of T cells resulted in FoxA1-mediated transcription of the transmembrane signalling protein PDL1 (ref. 3). The engagement of PDL1 on T cells with its receptor, programmed death 1 on APCs has been shown to induce anti-proliferative effects[32]. Here we identified that PDL1 is essential for neurons to bind to PD-1 on T$_{enc}$ cells and convert them to FoxA1$^+$T$_{regs}$. IFNβ is also produced by CNS-resident glial cells[14,33], which during neuroinflammatory process in EAE also express PDL1 (refs 34–36). Hence, this posts the question of why, unlike neurons, glial cells were incapable of converting T$_{enc}$ cells to FoxA1$^+$T$_{regs}$. It is plausible that these immunoregulatory capabilities of neurons are attributed to the fact that neurons, in contrast to glial cells, do not express major histocompatibility complex class II nor produce the predominant proinflammatory cytokines[7,37,38], and therefore incapable of conducting a full-activation signal in T cells, but rather exert tolerogenic signal and hence cause a different outcome for T$_{enc}$ cells. Understanding these

**Figure 4 | Generation of FoxA1$^+$T$_{reg}$ cells requires neuronal FoxA1 and PDL1.** (**a**) Percentage of FoxA1$^+$T$_{regs}$ upon co-culture of T$_{encs}$ with CGNs, with or without transwell. Graphs are mean ± s.e.m., $n = 3$. One-way analysis of variance (ANOVA) test was used, ***$P < 0.001$. (**b**) Fold change of *Foxa1* mRNA and (**c**) *Pdl1* mRNA in CGNs treated with or without rIFNβ (100 U ml$^{-1}$). Graphs are mean ± s.e.m., $n = 3$, two-way ANOVA test was used, *$P < 0.05$ and **$P < 0.01$. (**d**) *Foxa1* and (**e**) *Pdl1* mRNA in CGNs (left) and quantification (right). Graphs are mean ± s.e.m., $n = 3$, Student's $t$-test was used, ***$P < 0.001$. (**f**) Cytosolic (Cyto) and nucleus (Nu) fractions from CN (left). Graph shows normalized optical density (IOD) ratios between cytosolic/nucleus fractions of quantified FoxA1 protein bands (right). Graphs are mean ± s.e.m., $n = 2$–4. Non-parametric Mann–Whitney test was used for comparison of treated groups with control group. *$P < 0.05$ and **$P < 0.01$. (**g**) CGNs with or without treatment with rIFNβ. NF200 (green), FoxA1 (red) and DAPI (blue). Scale bars, 5 μm. (**h**) WB of FoxA1, PDL1 and vinculin in CNs extracts with or without treatment with rIFNβ. (**i**) Representative FACS histogram of neuronal FoxA1 and PDL1 after *Foxa1* siRNAKD in CGNs and (**j**) WB of FoxA1, PDL1 and vinculin after *Foxa1* siRNAKD in CGNs. (**k**) Percentage of FoxA1$^+$T$_{regs}$ after co-culture of T$_{encs}$ with CGNs after *Foxa1* or control siRNAKD in CGNs. Graphs are mean ± s.e.m., $n = 3$, Student's $t$-test was used, ***$P < 0.001$. (**l**) Neuronal PDL1 after *Pdl1* or UNC (Universal Negative Control) siRNAKD in CGNs. (**m**) Percentage of FoxA1$^+$T$_{regs}$ upon co-culture of T$_{encs}$ with CGNs after PDL1 or UNC siRNAKD in *Ifnb$^{+/+}$* CGNs. Graphs are mean ± s.e.m., $n = 3$, Student's $t$-test was used, ***$P < 0.001$. (**n**) Percentage of FoxA1$^+$T$_{regs}$ upon co-culture of T$_{encs}$ with CGNs after PDL1 was blocked in *Ifnb$^{+/+}$* neurons utilizing anti-PDL1 or PD1 was blocked in T$_{encs}$ utilizing anti-PD1 antibodies. Graphs are mean ± s.e.m., $n = 3$, one-way ANOVA test was used, ***$P < 0.001$. (**o**) Neuronal PDL1 after transfection of pIRES2-EGFP-PDL1 or pIRES2-EGFP in CGNs. (**p**) Percentage of FoxA1$^+$T$_{regs}$ upon co-culture of T$_{encs}$ with overexpression of PDL1 in *Ifnb$^{-/-}$* CGNs. Graphs are mean ± s.e.m., $n = 3$, one-way ANOVA test was used, ***$P < 0.001$.

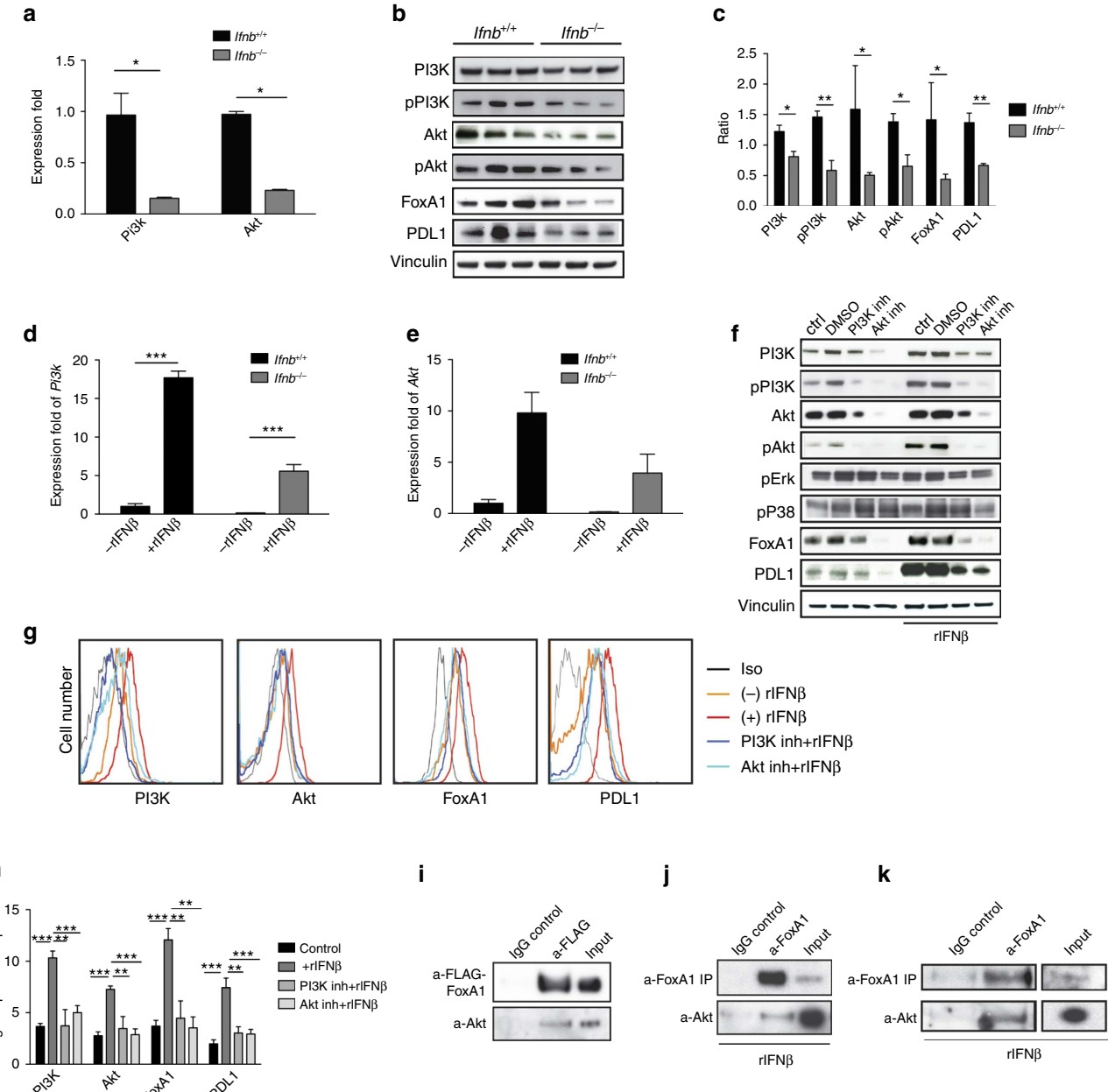

**Figure 5 | Activation of PI3K/Akt signalling regulates neuronal FoxA1 and PDL1.** (**a**) Real-time PCR analysed fold changes of *Pi3k* and *Akt* mRNA in CGN. Graphs are mean ± s.e.m. from three independent experiments. Two-way analysis of variance (ANOVA) test was used, \*P < 0.05. (**b**) WB of CGNs from 3 different individual *Ifnb*<sup>+/+</sup> and *Ifnb*<sup>−/−</sup> mice and (**c**) quantifications of proteins to vinculin. Western blottings from **b** were quantified with ImageJ, then ratio was calculated by using quantification of each individual band/control (vinculin). Graphs are mean ± s.e.m., *n* = 3 independent experiments. Two-way ANOVA test was used, \*P < 0.05 and \*\*P < 0.01. (**d**) *Pi3k* and (**e**) *Akt* mRNA fold changes obtained by real time PCR in CGNs with and without rIFNβ treatment (normalized to GAPDH, as housekeeping gene). Graphs are mean ± s.e.m. from three independent experiments. Two-way ANOVA test was used, \*\*\*P < 0.001. (**f–h**) Effects of PI3K inhibitor (Wortmannin in DMSO, 200 nm) and Akt inhibitor (MK-2206 in DMSO, 1 μM) in the presence or absence of rIFNβ (100 U ml<sup>−1</sup>) in CGNs. PI3K, pPI3K, Akt, pAkt, pErk, pP38, FoxA1 and PDL1 expressions by WB (**f**). (**g**) Representative FACS of PI3K, Akt, FoxA1 and PDL1 expressions in CGNs and (**h**) quantification of protein expression by FACS. Graphs are mean ± s.e.m., *n* = 3. Two-way ANOVA test was used, \*\*P < 0.01 and \*\*\*P < 0.001. (**i**) Co-IP of Akt with FLAG-tagged FoxA1 after overexpression of FoxA1-FLAG pCDNA3.1 in N2A cells. (**j**) Co-IP of Akt with rIFNβ-induced endogenous FoxA1 in N2A cells. (**k**) Co-IP of Akt with rIFNβ-induced endogenous FoxA1 in primary neurons.

pathways and the possible defects are imperative for designing future therapeutic approaches for chronic and progressive neuroinflammatory conditions such as progressive MS.

## Methods

**Mice.** *Ifnb*<sup>−/−</sup> mice were backcrossed to C57BL6 or C57BL/10.RIII strains of mice over 20 generations[3]. The WT littermates were used (*Ifnb*<sup>+/+</sup>) as controls. Mice were purchased from The Jackson Laboratroy, USA. The mice were bred

and kept at conventional animal facilities at the University of Copenhagen. One- to 7-day-old newborn pups were used in neuron-culture experiments. Eight- to 12-week-old *Ifnb*<sup>−/−</sup> and WT mice[3] or *nes*<sup>Cre</sup>:*Ifnar*<sup>fl/fl</sup> (Ifnar gene targeting only in neuroectodermal cells) and their WT littermates (*Ifnar*<sup>fl/fl</sup>) were used for inducing EAE[10].

All experiments were performed in accordance with the ethical committees in Copenhagen, Denmark and approved by the respective Institutional Review Boards (2013-25-2934-00807). Regarding the age and gender, mice were equally allocated to experimental groups.

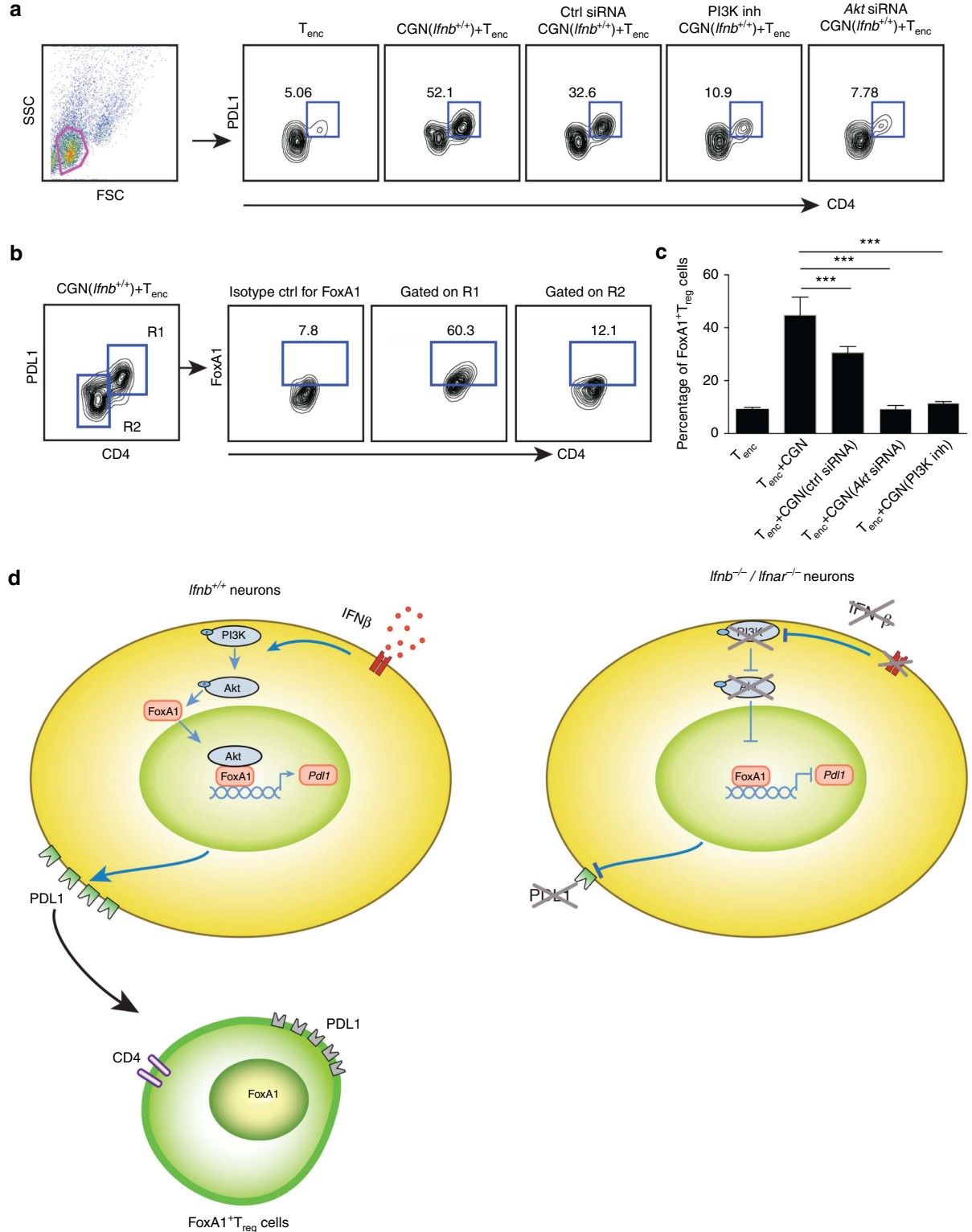

**Figure 6 | Neuronal ability to generate FoxA1$^+$T$_{regs}$ is IFNβ-mediated PI3K-Akt-FoxA1 and PDL1 dependent.** (**a**) Representative FACS plots of T$_{enc}$ cells after co-culture with CGNs, with and without *Akt* siRNA KD or PI3K inhibitor (Wortmannin in DMSO, 200 nm). FoxA1$^+$T$_{reg}$ cells were gated on CD4$^{high}$PDL1$^{high}$ cells. (**b**) FoxA1 expression was gated in CD4$^{high}$PDL1$^{high}$ cells (R1) and in CD4$^{low}$PDL1$^{low}$ cells (R2) and (**c**) quantified percentage of FoxA1$^+$T$_{regs}$. Graphs are mean ± s.e.m., $n = 3$, one-way analysis of variance (ANOVA) test was used, ***$P < 0.001$. (**d**) Schematic drawing of how neuronal endogenous IFNβ signalling leads to PI3K and Akt phosphorylation. Total and phosphorylated (p)Akt binds to FoxA1, FoxA1-Akt complex translocates to nucleus and binds to *Pdl1* promoter, consequently resulting in PDL1 expression. This signal is essential for neurons to convert T$_{enc}$ cells to anti-inflammatory FoxA1$^+$T$_{reg}$ cells. In contrary, lack of endogenous IFNβ signalling in *Ifnb*$^{-/-}$ neurons leads to insufficient phosphorylated PI3/Akt signalling, defective FoxA1 mediated PDL1 expression and thereby inability to generate FoxA1$^+$T$_{reg}$ cells.

**Induction of neuroinflammation EAE.** To establish neuroinflammation in the CNS, active or adoptive EAE[14] was induced in $Ifnb^{-/-}$ or their WT mice or in $nes^{Cre}:Ifnar^{fl/fl}$ (ref. 10). Active EAE was induced with $MBP_{89-101}$ or $MOG_{35-55}$, respectively[10,14]. Each animal was subcutaneously immunized in the base of the tail with 100 μl of a 1:1 emulsion of 150 μg of $MOG_{35-55}$ or 250 μg of $MPB_{89-101}$ in PBS and complete Freund's adjuvant (CFA) containing 500 μg of *Mycobacterium tuberculosis* H37Ra (Difco). An intraperitoneal injection of 500 ng of pertussis toxin/PT (*Bordetella pertussis*; Sigma-Aldrich) dissolved in 100 μl of PBS was given on the day of immunization (day 0). Mice were observed for clinical signs of EAE every day. For adoptive transfer EAE, mice were irradiated (500 rad) and injected in the tail vein with a cell suspension of $MBP_{89-101}$-specific T cells. At day 0 and 2, each animal was given an intraperitoneal injection of 500 ng of pertussis toxin. The reason for utilizing two different models of EAE either in C57BL/6 or C57BL/10RIII backgrounds was to address both chronic EAE induced by $MOG_{35-55}$, which is commonly used because of the availability of many transgenic strains in the C57BL/6 genetic background, and to utilize a relapsing-remitting EAE model induced by $MBP_{89-101}$, which is the model for relapsing–remitting MS in C57BL/10RIII background[10,14]. In addition, such an approach excludes the possible bias introduced by slight differences in the genetic backgrounds (C57BL/6 versus C57BL/10RIII) rather than the function of the target genes, here *Ifnb* and *Ifnar*.

Clinical scoring for EAE was as follows: 0, no disease; 1, limp tail; 2, limp tail and ataxia, as well as hind limb weakness/unsteady walk; 3, both hind limbs affected; 4, complete paralysis of hind limbs until hips; and 5, moribund or dead.

**DTH response.** Mice aged 8–15 weeks were immunized with 250 μg of $MBP_{89-101}$ emulsified in 50 μl of PBS and 50 μl of CFA. At day 13 p.i., mice were injected with 100 μg of $MBP_{89-101}$ (in PBS) + $FoxA1^{+}T_{regs}$ ($3 \times 10^{4}$ cells per ear) in the right ear or 100 μg of $MBP_{89-101}$ + control T cells ($3 \times 10^{4}$ cells per ear) in the left ear. Control mice received an injection of 100 μg of $MBP_{89-101}$ in the left ear and PBS + control T cells ($3 \times 10^{4}$ cells per ear) in the right ear. DTH response was measured as the difference in thickness (mm) of the right and left ears after 24 and 48 h, respectively. Data for the control T-cell group are presented as: (ear thickness after injection with $MBP_{89-101}$ + control T cells) − (ear thickness after injection with $MBP_{89-101}$). Data for the $FoxA1^{+}T_{regs}$-treated groups are presented as: (ear thickness injected with $MBP_{89-101}$ + $FoxA1^{+}T_{regs}$) − (ear thickness after injection with $MBP_{89-101}$).

**Neuronal cultures.** Two or three mice per group were pooled for each independent experiment. Neurobasal medium (ThermoFisher, 21103049) was supplemented with 2% B27 (B-27 supplement, serum free, ThermoFisher, 17504044), 1% Penicillin–Streptomycin and 125 μl glutamine (200 mM, ThermoFisher, 25030081). Culture plates with 96 or 24 wells were coated with poly-D-lysine (70 μg ml$^{-1}$, P7280, Sigma-Aldrich) in sterile distilled water, incubated for at least 30 min and washed with sterile distilled water.

CGNs cultures were obtained by decapitating a 7-day-old mouse, whereas CNs were dissected from 1-day-old pup brains. The brains were placed immediately in a 50 ml tube with Hank's balanced salt solution (HBSS) on ice, before dissecting in a Petri-dish containing 1 ml HBSS in a sterile hood. The cerebellum and cortex were dissected out and kept intact, while meninges were removed. A single-cell suspension was prepared by adding trypsin to a final concentration of 200 μg ml$^{-1}$. DNAse and FCS were added to final concentrations of 0.12% and 0.5%, respectively, before incubation for 6 min. Finally, the tissue was mechanically dissociated to single cells using a fire-constricted Pasteur pipette and cells were seeded at a concentration of $4 \times 10^{5}$ cells per ml. CGNs were cultured for 3–5 days and CNs for 21 days before co-culture with T cells.

**Immunofluorescence staining of neurons.** CGNs from $Ifnb^{+/+}$ and $Ifnb^{-/-}$ mice were cultured 3 days *in vitro* on eight-well LabTek chamber slides (Nunc). Cells were fixed in 4% paraformaldehyde, permeabilized with 0.2% TritonX-100 and stained with NF-200 (N4142, 1:150, Sigma-Aldrich) and mouse anti-human (mouse) FoxA1 antibody (2F83, 1:200 Millipore). Hoechst 33342 (H3570, 1:2,000, Invitrogen) was used for staining nuclei. For measuring the neuronal culture purity, anti-glial fibrillary acidic protein (GFAP) (13-0300, 1:800, Invitrogen) and anti-β III tubulin (ab52901-100, 1:100, Abcam) were used to stain the cells and then positive cells were counted in at least ten randomly selected areas per culture in three independent experiments.

**Glial cell culture.** Primary mixed glial cell cultures were established from the forebrain and cerebellum of 1- to 2-day-old B10.RIII ($Ifnb^{+/+}$) mice. The tissues were carefully dissected out and freed of meninges before being placed in HBSS solution supplemented with 1 mM pyruvate and 11 mM glucose. Tissue was then chopped into smaller pieces using a razor blade and incubated with 1% trypsin (Sigma-Aldrich) for 10 min at 37 °C. Thereafter, DNase and FCS were added to a final concentration of 0.12% and 0.5%, respectively, and incubated for 8 min. Finally, the solution was mechanically dissociated to single cells using a fire-constricted Pasteur pipette. The cells were seeded at a concentration of $1.5 \times 10^{5}$ cells per cm$^{2}$ in a 1:1 mix of DMEM and F12 medium (Invitrogen Life Technologies) supplemented with 0.16 μM ml$^{-1}$ penicillin, 0.03 μM ml$^{-1}$

streptomycin and 5% FCS. Medium was changed every third to fourth day. After 7 days in culture, oligodendrocytes were detached from the cell monolayer by shaking on an orbital shaker. After two to three passages, which initially consisted of a mixture of 20–30% Mac-1$^{+}$ microglia and 70–80% GFAP$^{+}$ astrocytes, these cultures were further enriched. After 8 days in culture, the mixed glial cells were washed and vigorously shaken at 900–1,000 r.p.m. for 3 h on an orbital shaker. For microglial cultures, the floating cells were collected, washed and reseeded in 96-well plates (Nunc) at a concentration of $1 \times 10^{4}$ cells per well. After adhering overnight, nonadherent or loosely attached cells were washed away. The adherent cells represented >97% pure microglial cells as determined by Mac-1$^{+}$ staining, with <3% being GFAP$^{+}$. For astrocyte cultures, the still adherent cells were trypsinized and reseeded in flasks and left to adhere for 30 min. Floating or loosely attached cells were recovered by mild shaking by hand and the adhesion for 1 h. Cells in the supernatant were thereafter collected, washed, and reseeded in 96-well plates (Nunc) at a concentration of $4 \times 10^{4}$ cells per well. These cells were >96% pure astrocytes as determined by GFAP$^{+}$ staining, <4% were Mac-1$^{+}$ cells. Both these cultures were then used for coculture with T cells.

**Establishment of $T_{enc}$ cell lines.** To generate T-cell lines, 8- to 12-week-old male B10RIII ($Ifnb^{+/+}$) were immunized in the flank and tail base with 200 μl of a 1:1 emulsion of 250 μg of $MBP_{89-101}$ in PBS and CFA containing *M. tuberculosis* H37Ra (Difco). $MOG_{35-55}$ T-cell line was generated by immunized with C57BL6 mice with 150 μg of $MOG_{35-55}$. Draining lymph nodes were collected 10 days after immunization and a single-cell suspension prepared in PBS by passing through a sieve. Cells were washed and suspended in DMEM with Glutamax-1 (Gibco) supplemented with 10 mM HEPES buffer, 10% heat-inactivated FCS (Sigma), 100 U ml$^{-1}$ of penicillin, 100 μg ml$^{-1}$ of streptomycin and 50 μM 2-mercaptoethanol, to make complete DMEM (cDMEM). T cells were cultured as $5 \times 10^{5}$ cells in round-bottomed 96-well culture plates (Nunc) in 200 μl of cDMEM, in a humidified 37 °C atmosphere containing 5% CO$_2$. T cells were stimulated for 4 days with 50 μg of $MBP_{89-101}$. After a resting phase of 8 days in media supplemented with 800 pg ml$^{-1}$ of interleukin (IL)-2 (obtained from the supernatant of an IL-2 transfected X63 cell line), T cells were re-stimulated with 20 μg of peptide and irradiated APCs. APC were generated from spleen cells of syngeneic mice, prepared as described above for lymph nodes with an additional 0.84% NH$_4$Cl treatment to lyse red blood cells, and irradiated with 3,000 rad before being used at a concentration ten times higher than the T cells. To expand highly specific T-cell lines, stimulation was repeated at intervals of 10–30 days. Between stimulations, the antigen-containing media was removed and the T cells kept in a resting state in cDMEM supplied with IL-2. The media was changed every fourth day. In all experiments, T-cell lines had gone through a total of four to ten stimulation rounds.

**Co-culture of neurons with T cells.** CGNs and/or CNs were prepared[7,8] and seeded at $4 \times 10^{5}$ cells per ml in 96- or 24-well plates with neuronal media for 3 and 21 days, respectively. T-cell lines were re-stimulated with antigen and APCs for 48 h and are referred to as activated $T_{enc}$ cells. Neurons and activated syngeneic $T_{enc}$ cells were washed twice and co-cultured at a 1:1 ratio for 24 h, unless stated otherwise. $MOG_{35-55}$ T-cell line and $MBP_{89-101}$T-cell line were co-cultured with neurons from C57BL6 mice and C57BL/10.RIII mice, respectively.

For some experiments, PI3K inhibitor Wortmannin (catalogue number S2758, Selleckchem), Akt inhibitor MK-2206 (catalogue number S1078, Selleckchem), rIFNβ (12405-1, PBL Assay Science) and anti-PDL1 (10 μg ml$^{-1}$, Anti-Mouse CD274/B7-H1) Functional Grade Purified, 16-5982-82, eBioscience) were added to the CGNs. Anti-PD1 (10 μg ml$^{-1}$, Anti-Mouse CD279/PD-1, 16-9985-82, eBioscience) was added to T cells before co-cultures.

**Quantitative real-time PCR.** Total RNA was isolated using a QIAGEN kit (QIAGEN), reverse transcribed into complementary DNA, and amplified and quantified by SYBR Green (Bio-Rad) detection. Relative mRNA expression was calculated using *GAPDH* gene expression as an endogenous reference (PPM02946E). The primers were purchased from SAbiosciences (Qiagen) as following: *Akt1* (PPM03377G), *Foxa1* (PPM04764H), *Pdl1* (PPM34637A) and *Pi3k* (PPM03374G).

**FACS staining and sorting using FACSAria.** After washing in FACS buffer (2% FCS in PBS), cells were incubated with anti-Fc receptor Ab (24.G.2, our hybridoma collection) at 10 μg ml$^{-1}$. Thereafter, cells were incubated with biotinylated, fluorescein isothiocyanate (FITC)- or phycoerythrin (PE)-labelled antibodies. For intracellular staining, cells were fixed and permeabilized using BD Cytofix/Cytoperm or using fixation and permeabilization solutions from Human Treg Flow Kit. All antibodies were used at 1–5 μg ml$^{-1}$ and were allowed to bind for 20 min on ice. The antibodies used were as follows: APC or FITC-anti-mouse CD4 (L3T4, 1:200, BD), FITC or biotin-anti-mouse TCR (553169, 1:200, BD), PE-anti-mouse PDL1 (MIH5, 1:200, BD), isotype control for PDL1: PE-rat IgG2a isotype control (eBR2a, 1:200, 8012-4321-025, eBioscience), mouse anti-human (mouse) FoxA1 antibody (2F83, 1:200, Abcam, catalogue number ab40868) and secondary Alexa Fluor 488 goat anti-mouse (A-11029, 1:1,000, Invitrogen) or Alexa Fluor 647 goat anti-mouse (558865, 1:1,000,

**Table 1 | ON-TARGETplus SMARTpool siRNA sequences.**

| Gene | Catalogue number | Target sequence (5′–3′) | Antisense (5′–3′) |
|---|---|---|---|
| Cd274(Pdl1) | A-040760-13 | UGAGCAUGAACUAAUAUGU | ACAUAUUAGUUCAUGCUCA |
| | A-040760-14 | GUAUCAGCUCUCAGAUUUC | GAAAUCUGAGAGCUGAUAC |
| | A-040760-15 | UCUGUAGACACCAUUUAUA | UAUAAAUGGUGUCUACAGA |
| | A-040760-16 | UCAUGGUGUUGGAUUGGUG | CACCAAUCCAACACCAUGA |
| Akt1 | A-040709-14 | GAUUCAUGUAGAAAACUAU | AUAGUUUUCUACAUGAAUC |
| | A-040709-15 | CGUGUGACCAUGAACGAGU | ACUCGUUCAUGGUCACACG |
| | A-040709-16 | GCGUGGACCAUGUACGAGAU | AUCUCGUACAUGGUCCACGC |
| | A-040709-17 | UUCUUUGCCAACAUCGUGU | ACACGAUGUUGGCAAAGAA |
| Foxa1 | A-046238-13 | CGGACGUCCUUAAGUGAAA | UUUCACUUAAGGACGUCCG |
| | A-046238-14 | CUAUGGACUUAAUAUCAUG | CAUGAUAUUAAGUCCAUAG |
| | A-046238-15 | GGCUCAUCCAGUGUUAAUG | CAUUAACACUGGAUGAGCC |
| | A-046238-16 | GACUGUUACUUUAUUAUUG | CAAUAAUAAAGUAACAGUC |

Invitrogen), isotype control for FoxA1: purified mouse IgG1κ isotype (MOPC-21, 1:100, BD). Violet-fluorescent reactive dye (L34955, 1:1,000, Invitrogen) was used for LIVE-DEAD marker.

Cells were acquired with an FACSAria (BD Biosciences) using the FACSDiva software for acquisition after exclusion of duplets. Dead cells were discriminated in all staining using the LIVE/DEAD Fixable Dead Cell Stain Kit for 405 nm excitation (L34955, Invitrogen). FlowJo 8.8.6 (Tree Star) was used for further analysis.

FoxA1$^+$ $T_{reg}$ cells were gated by CD4$^{high}$PDL1$^{high}$ and FoxA1$^+$ expression.

**siRNA silencing.** Accell SMART pool siRNA was purchased from Dharmacon (catalogue number E-046238-00, Thermo Scientific) and was introduced into neurons according to the manufacturer's protocol. Briefly, SMART pool siRNA combines four different siRNAs to reduce off-target effects. The Accell siRNA is also designed for optimal delivery to hard-to-transfect cells and no transfection reagents were required to introduce the siRNAs. The following SMART pool Accell siRNAs from GE Healthcare were used: Foxa1 (E-046238-00-0005), Cd274 (E-040760-00-0005) and Akt1 (E-040709-00-0005); for detail siRNA sequences, please see Table 1.

**Western blotting.** Proteins were extracted in 300 μl RIPA buffer (Sigma-Aldrich, R0278) from 3 × 10$^6$ CGNs, 15 μl of protein lysate was loaded on 4–12% SDS–PAGE gels and proteins were blotted onto polyvinylidene difluoride membranes. The membranes were blocked in 5% skim milk (Sigma-Aldrich, 70166) or 5% BSA in PBS-Tween-20 (0.05%) 1 h room temperature and incubated with: 1 h room temperature and then with horseradish peroxidase-conjugated secondary antibodies. The blots were developed using the ECL technique (Millipore, MA, USA).

The following antibodies were utilized: anti-PI3K (4257, Cell Signaling), anti-PI3K p85 (phospho Y607) antibody (ab182651, 1:1,000, Abcam), anti-Akt (9272, 1:1,000, Cell Signaling), anti-Akt (Phospho-Ser473) (4060L, 1:1,000, Cell Signaling), anti-phospho P38 (T180/Y182) (9211, 1:1,000, Cell Signaling), anti-phospho ERK (T202/Y204) (9101, 1:1,000, Cell Signaling), anti-FoxA1 (ab23738, 1:1,000, Abcam) and anti-PDL1 (BAF1019, 1:5,000, R&D), and anti-vinculin (1:100,000, v-9313, Sigma-Aldrich).

In some experiments, CNs were cultured for 7 days and stimulated with rIFNβ for 30, 60 or 120 min. Subsequently, the CNs were fractionated into a cytosolic and nucleus fractions[39]. Fractions were loaded on 4–12% gels according to cytosolic protein quantification determined by Pierce BCA protein assay kit. The protein concentrations loaded in the cytosolic fractions were around three times higher than in the nucleus fraction. Antibodies used were as follows: Polyclonal rabbit anti-FoxA1 (1:1,000, Abcam catalogue number ab23738), monoclonal mouse anti GAPDH (1:20.000, Abcam catalogue number ab9484) and polyclonal rabbit anti TBP (1:500, Santa Cruz catalogue number sc-204) were used.

Images have been cropped for presentation. Full-size images are presented in Supplementary Figs 1–4).

**Amaxa transfection of Pdl1 siRNA or pIRESII PDL1.** Neurons (5 × 10$^6$) from B10RIII Ifnb$^{+/+}$ mice were transfected with 10 μM of a Pdl1 siRNA (SASI_Mm02_00326460, MISSION siRNA, Sigma-Aldrich) or a control siRNA UNC-1 (universal negative control-1) (Sigma-Aldrich) using the Amaxa nucleo-fector technique (programme G13) according to the manufacturer's description (Mouse Neuron Nucleofector Kit, VPG-1001, Lonza, Basel, Switzerland). Ifnb$^{-/-}$ mouse neurons (5 × 10$^6$) were transfected with 10 μg pIRES2-EGFP-PDL1 or pIRES2-EGFP control using the Amaxa nucleofector technique (program G13) according to the manufacturer's instructions (Clontech). After 72 h, neurons were co-cultured with activated $T_{enc}$ cells or cultured alone in 96-well plates. Cells were then collected, stained and were analysed by FACS analysis. To

evaluate transfection efficiency, some of the neurons were stained with PE-anti-PDL1 (1:200, MIH5, BD) for FACS analysis.

**Co-immunoprecipitation.** N2A cells (Sigma-Aldrich) were lipofectamine transfected (Lipofectamine 2000, Invitrogen, Opti-MEM 1X Reduced-Serum Medium, GIBCO by Life Technologies) according to the manufacturer's protocol with FoxA1-FLAG pCDNA3.1 (2 μg DNA per ml) for FoxA1-FLAG IP. Cells were harvested 48 h after transfection in NP-40 1% lysis buffer (NaCl 150 mM, NP-40 1.0%, TRIS pH 8, 50 mM) supplemented with phosphatase and protease inhibitors.

For the IP of endogenous FoxA1, N2A cells or primary isolated neurons were treated with rIFNβ (100 U ml$^{-1}$) for 2 h before lysis in a minimum volume of RIPA buffer (R0278, Sigma) with protease and phosphatase inhibitors. The RIPA lysates were supplemented with NP-40 1% lysis buffer for the IP procedure.

The lysates were precleared with Sepharose protein G beads (Protein G Sepharose 4 Fast Flow, GE Healthcare) before incubation with capture antibody (4 μg ml$^{-1}$) (monoclonal α-FLAG M2, F1804-50UG, Sigma, its isotype control mouse IgG1, κ, 554121, BD Pharmingen, and α-FoxA1, ab5089, Abcam, and its isotype control goat IgG, ab37373, Abcam) on rotation at 4 °C overnight. The IP lysates were incubated with 50 μl sepharose beads per ml lysate on rotation for 1 h at 4 °C, followed by 3 × 5 min washing in NP-40 1% lysis buffer with protease and phosphatase inhibitors. Proteins were eluted from the beads in 100 μl 2 × LDS-load (NuPAGE LDS Sample Buffer (4 ×), Novex by Life Technologies) with 100 mM dithiothreitol by incubation at 70 °C for 10 min followed by spin down of the beads. Samples were loaded on a 4–12% Bis-Tris gel and transferred to a nitrocellulose membrane. Membranes were blocked for 1 h in 5% skimmed milk (Sigma-Aldrich, 70166) TBS-T (0.05% Tween20) and immunoblotted for the presence of IP FoxA1-FLAG or endogenous FoxA1 and co-IP endogenous AKT with specific antibodies monoclonal α-FLAG M2-Peroxidase (horseradish peroxidase) (1:8,000, A8592-.2MG, Sigma-Aldrich), α-FoxA1 (1:1,000, Ab23738, Abcam), α-Akt (1:15.000, C67E7 Rabbit monoclonal antibody 4691 Cell Signaling), α-Akt (1:5,000, 9272 Cell Signaling) and α-phospho-Akt (1:6,000, Ser473 4060 Cell Signaling). The blots were developed according to the Millipore ECL protocol.

**Histology and immunofluorescence histochemistry.** Brains and spinal cords of mice with EAE were dissected. Mice were either perfused and tissues fixed in 4% paraformaldehyde (PFA) and paraffin embedded, or tissues were embedded in OTC compound (Sakura Finetek Denmark ApS, Værløse, Denmark) and snap-frozen in isopentane on dry ice. Tissues were sectioned in 6–10 μm slices, were fixed in 4% Paraformaldehyd (PFA) for 10 min, either stained with haematoxylin and eosin (H&E, Histolab) staining or different antibodies were used for staining and visualized. Primary antibodies used for immunofluorescence (IF) were as follows: rabbit anti-TCR alpha (Abcam, ab18861, 1:30), rat anti-PDL1 (Abcam, ab80276, 1:200), mouse anti-FoxA1 (Millipore, 05-1466, 1:2,000) and rabbit anti-NF200 (Sigma, N4142, 1:50). Secondary antibodies to IgG used were as follows: goat anti mouse-Af568 (Invitrogen, A11031), goat anti-rat IgG- Af488 (Invitrogen, A11006), goat anti-rabbit IgG- Af633 (Invitrogen, A21071), goat anti-rabbit IgG-Pacific Blue (Invitrogen, P10994), all in concentration of 1:500, Drag5-633 (BioStatus, DR05500, 5 μm) or DAPI-Pacific blue (Invitrogen, D3571, 1:30,000) were used for nuclei visualization. IF images were taken with a Zeiss LSM510 confocal scanning microscope or with IN Cell Analyzer 2200 automated microscope. H&E images were taken with a NanoZoomer 2.0-HT digital slide scanner or Olympus BX51 microscope.

**Image quantification.** Automatic counting: IF images were taken with the IN Cell Analyzer 2200 automated microscope. Images were processed and quantified with ImageJ (Fiji version) and IN Cell Investigator software. The cells were identified as TCR or NF200 positive based on negative controls and then

intensities in the other separate channels were measured and scored as positive or negative using the automated software analysis.

For Infiltrating cells count, H&E-stained images were converted from NDPI version (NanoZoomer) to TIFF format with 'NDPI tools' plugin on ImageJ. Subsequently, the infiltrated areas were manually detected and a customized macro for counting cells was run for automatic cells count.

Manual counting: For some IF staining images that were manually quantified ImageJ was utilized. Negative controls were utilized as reference. DAPI channel was selected with the ImageJ function *find maxima* to automatically count the total number of nucleus (cells). Next, visualizing only DAPI and far red channel, TCR either NF200 (far red channel) were manually counted with ImageJ manual counting function, and then, co-expression with red channel (FoxA1) for the double positive and green channel (PDL1) for triple positive were evaluated.

**Statistical evaluations.** Statistical evaluation was performed using GraphPad Prism (GraphPad Software Inc.). Student's *t*-test was used for two groups comparison, one-way analysis of variance with post Tukey's multiple comparisons test and two-way analysis of variance with post Sidak's multiple comparisons test were used for more than two groups comparison. Non-parametric Mann–Whitney test was used for EAE score and IF staining quantification data. A value of $P < 0.05$ was considered significant.

**Data availability.** All relevant data are available from the authors on request.

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

## Acknowledgements

Support was received from Danish Council For Independent Research (DFF)-Medical Science (DFF-4183-00427B), Danish Multiple Sclerosis Society, Foundation for Research in Neurology, Ejnar Johnsen og Hustru's Mindelegat Fond and The Lundbeck Foundation.

## Author contributions

Y.L., A.M., P.E. and L.M.R. did experiments and analysed and prepared data. S.I.-N. designed and supervised the study, analysed and interpreted data. M.P. provided material from Nes^Cre:Ifnar^fl/fl. Y.L. and S.I.-N. wrote the manuscript. All authors read and contributed to the final manuscript.

## Additional information

**Competing financial interests:** The authors declare no competing financial interests.

