## [Peer Review File · Nature Communications]

Reviewers' comments:

Reviewer #1 (Remarks to the Author):

This study shows that the IFNAR signaling in the neurons enhances PD-L1 expression through FoxA1 induction. Cell-cell contact between a neuron and a T cell via PD-L1 on neurons induces FoxA1+PD-L1 positive T cells.

Regulation of encephalitogenic T cells by neurons is a significant and intriguing topic. However, this manuscript has at least several major issues. First, although IFN β is knocked down, there are still IFN α detected by IFNAR. It is not clear how IFN α is (or is not?) involved in many experimental settings in this study. Second, major conclusions were often drawn only from immunofluorescence (IF) panels without any quantitative evaluation. Third, do T cells used in the experiments express PD-1? In other words, it was not shown how T cells detect PD-L1 on neurons. Fourth, critical controls and flow gating are often missing. Fifth, figures are often poorly exhibited. And, finally, the study end up to be descriptive. Detailed comments are shown below.

- EAE severity data with scores has to be shown.
- The readout of FoxA1+Tregs is only based on the expression on FoxA1 and PD-L1. Do they truly show regulatory effects in vivo and ex vivo? This may have been shown in other articles, but the functional characterization of FoxA1+Tregs "in this study" is necessary.
- Does the induction of FoxA1+Tregs depend on PD-L1 on neurons? Not only the PD-1 expression T cells are critical, PD-1 KO T cells need to be tested whether the absence of PD-1 on the T cell side abolishes the effect. The PD-1 KO mouse line is commercially available from JAX.
- Why were B10RIII mice as T cell donors with B6 recipients? What is the reason for this? Is it appropriate 'not' to use B6?
- The method describes, "incubation with 4 μ g capture antibody..." This is just one example, but absolute amounts of molecules/Abs clearly do not mean anything here. They have to be shown as concentrations.
- "Proteins were extracted in 40 μ L SDS loading buffer..." Not only is this a problem in describing in an absolute value again, but it is very strange to use SDS loading buffer to extract protein.
- Spell out "Tenc" in Abstract.
- Fig. 2a. The numbers of CNS-infiltrated cells in the figure are too low to be from active EAE mice. How and when (which day) the samples were obtained? H&E staining panels need to be arranged differently. Infiltrated cells in the photos are visible, but not easily. Is demyelination identified too?
- Again, just the IF data without any quantitative values cannot be used to draw conclusions. This is the case for the majority of Fig. 1 and Fig. 2, as well as Fig. 4f.
- Lack of controls to set up gating in analyzing flow data. Fig. 3b, c; 4e; 6a, b.
- How were cell numbers enumerated? Flow? Eye-count from staining? Fig. 1b, c; 2a, d, f; 3a, d, e; 4a.
- Fig. 4b, c; 5d, e. Do the levels of IFN β in the culture supernatants correspond to the results?
- The conclusion - enhanced FoxA1 nuclear translocation - only from the Fig. 4f is an overstatement. Western blotting with nuclei followed by quantitative evaluation is suggested.
- Fig. 4f and other IF panels with inset letters. The letters are too small.
- Fig. 4m. An empty vector control is required. Expression levels of PD-L1 have to be shown too.
- How the peaks were gated in Fig. 4h, k; 5g to consider some cells to be positive? Were the results in Fig. 4, j, l, m and Fig. 5h obtained from the histograms in Fig. 4h, k; 5g?
- Total Akt levels are low in Ifn β ^{-/-} samples in Fig. 5b. Simply not good data.
- Fig. 5i. IP has to be performed with CGNs. IgG IP control also shows background pAkt. Was this IP clean enough? Need better labeling in the figure; e.g., labeling with Ab names used for IP and WB will help.

- Method section. Spell out B27 and PS. Please also write which Ab was used for what, if it is not discussed in the main text, such as NF-200.
- Please show primer sequences for qPCR.
- Five % "milk" was used in a WB blocking solution. What exactly is "milk"?
- Where is the source of 3T3-L1 cells?
- Fig. 2a legend. What exactly are "infiltrating inflammatory cells"? It says CNS, but does it mean spinal cord of the brain?
- Labeling "e" is missing in Fig. 2e.
- Fig. 4j, l, m. How were the data obtained?
- Fig. 5c. What does it mean "ratio." How was it calculated?

Reviewer #2 (Remarks to the Author):

In a previous report, I-N et al reported that neurons instruct CD4+ T cells to differentiated to FoxA1+ Treg cells and that this instruction involves production and release of IFN β . The present report presents experiments to cement and extend the previous observations. It concludes that neurons act directly on invading effector T cells to divert them towards the FoxA1 direction.

This is an interesting concept but its formal proof has to be strengthened by additional data. So far, induction may occur indirectly, via glia, especially astrocytes.

First, shouldn't NesCre :Ifnar1/fl mice should lack IFNAR on neuron plus astrocytes? Second, the protocols used to generate neuronal cultures permits growth of glia cells, mostly astrocytes, thus an estimate of this contribution should be given. A GFAP staining of co-cultures should be shown at least. There is also a query related to the site of conversion in the CNS. Apparently, FoxA1+ T cells are differentiating and residing near neuronal bodies, placed in the grey matter, with very sparse infiltration. The perivascular areas of the white matter harboring the mass of inflammatory cells, are remote.

Another important issue: previous work by others pointed to the importance of neuronal electric activity. Would silencing of neurons (e.g. by tetrodotoxin) cancel induction? Most T cells invade the spinal cord and brain areas remote of neuronal bodies; how could they induce FoxA1 in T cells?

Less fundamental:

Fig.1 should be described in more detail: a) which brain region shown? How was infiltration quantified? Also give EAE grades. Line 75: "did NOT find apparent FoxA1+ Tregs" contradicts to Fig.1c with 20 vs 35% of such cells, and line 75. Where is "neuronal FoxA1" shown (Fig.1d is not helpful)? Neuronal bodies sit in spinal cord grey matter, which is not a predilection of T cells, see above.

Legend to Fig.4: (f) and (g) are interchanged. NF200 expression mostly in nucleus, why?

Forced expression of PDL-1 overrides IFN β deficiency. Why would PDL1 on microglia and other cells (T cells included) fail to substitute? Is there an additional neuronal factor at work?

We appreciate immensely the positive evaluation, valuable comments and suggestions by all reviewers, addressing which surely has improved and solidified our current manuscript and its message.

We are also happy to emphasize that we found all recommendations to the point and have basically addressed them all when suggested experimentally or in the text. Please see below specifications and detailed response by point-to-point addressing each point using blue color to indicate our response from the raised points by the reviewers.

Reviewers' comments:

Reviewer #1 (Remarks to the Author):

We appreciate very much the positive evaluation of our current manuscript by reviewer #1. We have taken the liberty to subdivide some of the comments and suggestions to easier address and discuss them below text in blue.

This study shows that the IFNAR signaling in the neurons enhances PD-L1 expression through FoxA1 induction. Cell-cell contact between a neuron and a T cell via PD-L1 on neurons induces FoxA1+PD-L1 positive T cells.

Regulation of encephalitogenic T cells by neurons is a significant and intriguing topic. However, this manuscript has at least several major issues.

First, although IFN β is knocked down, there are still IFN α detected by IFNAR. It is not clear how IFN α is (or is not?) involved in many experimental settings in this study.

As previously reported by others and also by us (Ejlervskov 2015 Cell), IFN β seems to be required even for IFN α production (Erlandsson, L., Blumenthal, R., Eloranta, M.L., Engel, H., Alm, G., Weiss, S., and Leanderson, T. 1998, Interferon-beta is required for interferon-alpha production in mouse fibroblasts. *Curr. Biol.* 8, 223–226.), we do not find compensatory function of IFN α in IFN β KO neurons (Ejlervskov 2015 Cell). Although under other conditions, IFN α might have additional effects if it would be regulated independent of IFN β , there is no evidence that is the case in neurons and this has not been the focus of our current investigation, never the less, because 14 IFN α alleles exist, it is not easy to address this issue. However, there are no much differences between IFN β KO and IFNAR KO mice or neurons in regard to the findings reported in our current study, which is arguing against a critical or differential effects of IFN α in the studied conditions. Therefore, nowhere in the manuscript we have referred to type I signaling for this purpose, but we refer to IFN β -IFNAR signaling, which is what we have studied.

Second, major conclusions were often drawn only from immunofluorescence (IF)

panels without any quantitative evaluation.

All are included, please see answers to the specifics below.

Third, do T cells used in the experiments express PD-1? In other words, it was not shown how T cells detect PD-L1 on neurons.

We have included new data (see figure 4n) showing that neuronal PDL1 and PD-1 on T cells are needed for FoxA1+Treg cell generation, if any of these two are blocked, neuronal induction of FoxA1+Treg cells are inhibited.

Fourth, critical controls and flow gating are often missing.

All are included, please see answers to specifics below.

Fifth, figures are often poorly exhibited. And, finally, the study end up to be descriptive. Detailed comments are shown below.

- EAE severity data with scores has to be shown.

This is included, please See figure 1a.

- The readout of FoxA1+Tregs is only based on the expression on FoxA1 and PD-L1. Do they truly show regulatory effects in vivo and ex vivo? This may have been shown in other articles, but the functional characterization of FoxA1+Tregs "in this study" is necessary.

As the reviewer is correctly referring, we have already shown both in vivo and in vitro suppressive potentials of neuron-induced FoxA1+Tregs extensively in Liu et al 2014 in Nature Medicine. Although we find it not necessary that each paper should repeat all the work of previous paper/s to be legitimate specially if it was the work of others, we of course needed to show that we also can reproduce others results, but these extensive experiments are done and published by us. However as suggested by the reviewer #1, we included again in figures 3d & 3e in vitro and in vivo suppressive capacity of neuron-induced FoxA1+Treg cells, respectively.

- Does the induction of FoxA1+Tregs depend on PD-L1 on neurons? Not only the PD-1 expression T cells are critical, PD-1 KO T cells need to be tested whether the absence of PD-1 on the T cell side abolishes the effect. The PD-1 KO mouse line is commercially available from JAX.

Please see above answer to this question in the third point. We included new experimental data in current figure 4n showing that both neuronal PDL1 and T-cell PD1 are required for this interaction by blocking these transmembrane ligand and receptor. However, although we did our best to get JAX mice, it has taken several months and the mice are not yet even delivered and since we have shown it conclusively by blocking this interaction, and since it is not the main point and focus of this manuscript, we are sure this will suffice.

- Why were B10RIII mice as T cell donors with B6 recipients? What is the reason for this? Is it appropriate 'not' to use B6?

We did not find where such a type error might have happened, but for clarification, both B10RIII and B6 are used as a source for primary neuronal cultures and for T cell line (MBP89-101, and MOG35-55 respectively) generation. To make it clear the M&M is reformulated to convey this message and that syngenic T cell lines are used together with neurons for the co-culture.

- The method describes, "incubation with 4 µg capture antibody..." This is just one example, but absolute amounts of molecules/Abs clearly do not mean anything here. They have to be shown as concentrations.

It has been changed to give concentration of antibodies and other reagents.

- "Proteins were extracted in 40 µL SDS loading buffer..." Not only is this a problem in describing in an absolute value again, but it is very strange to use SDS loading buffer to extract protein.

We have reformulated the section in M&M to properly describe the technique.

- Spell out "Tenc" in Abstract.

It is addressed.

- Fig. 2a. The numbers of CNS-infiltrated cells in the figure are too low to be from active EAE mice. How and when (which day) the samples were obtained?

The CNS tissues are dissected at day 35 post EAE. This was to address differences in the chronic phase of the disease, as we were interested in finding the defects associated with lack of suppression in the CNS and since our previous data suggested increased in FoxA1+Tregs after the acute phase has passed and mainly during stable/remission phase (Liu et al 2014 NM and fig 1 of the current manuscript). These are already addressed in detail both in M&M and results.

- H&E staining panels need to be arranged differently. Infiltrated cells in the photos are visible, but not easily. Is demyelination identified too?

It is not easy to guess what the reviewer's wish for "arrange differently", but since the reviewer refers to the visibility, we assumed larger images are preferred and therefore we increased the size. Regarding demyelination, yes these mice develop increased demyelination, but this is previously reported (Prinz et al. 2008, Immunity) and since it is not the focus here, it was not studied again.

- Again, just the IF data without any quantitative values cannot be used to

draw conclusions. This is the case for the majority of Fig. 1 and Fig. 2, as well as Fig. 4f.

Thanks for the remark, most of the IF were already quantified and shown, but as recommended the only missing one is now added to the current version (fig 1h) and new WB are performed to add to Fig 4f. Please see below for the details.

IF in figure 1, i.e. the percentage of TCR+ cells and FoxA1+TCR+PDL1+ (FoxA1+Tregs) were already quantified and shown (currently as fig 1c & 1d). IF shown in fig. 1e is a closed up of fig.1b, which was already quantified (current fig. 1c and 1d).

The reason for inclusion of fig. 1e (basically similar to fig 1b) is a closed up, to show that T cells expressing FoxA1 could be detected in the parenchyma in the vicinity of neuronal soma.

The IF in fig. 1f is just an overview of cerebellum which shows qualitative differences in neuronal PDL1 expression. However, the quantification of actual NF200+FoxA1+PDL1+ in spinal cords (same to all other quantifications) are shown in fig. 1h.

Fig 2. H&E i.e. the inflammation was already quantified and were shown and still in the current in Fig. 2a. While representatives H& E were shown (before 2a right) and currently in fig 2b. IF shown in Fig 2c is an overview of representative i.e. TCR+FoxA1+PDL1+ cells, which are also accompanied with other representatives shown in higher magnifications (Fig. 2d) and which were already quantified (current fig. 2e).

Fig 2f is showing NF200+FoxA1+PDL1+ in representative spinal cords and were already quantified (currently moved as fig. 2g).

Fig 4f is not meant to be any quantitative data, it is showing the localization differences. The quantitative data of nuclear vs. cytoplasmic FoxA1 are performed by WB as suggested by the reviewer #1 and are currently shown as Fig 4f.

- Lack of controls to set up gating in analyzing flow data. Fig. 3b, c; 4e; 6a, b.

The controls and gating strategy are included in all corresponding figures.

- How were cell numbers enumerated? Flow? Eye-count from staining? Fig. 1b, c; 2a, d, f; 3a, d, e; 4a.

The details are included in M&M.

- Fig. 4b, c; 5d, e. Do the levels of IFN β in the culture supernatants correspond to the results?

Fig 4b, 4c and 5d are neuronal cultures treated with same concentration of

recombinant (r)IFN β or without, so the levels of IFN β are corresponding to the treatment, endogenous levels in WT vs. lack of it in IFNBKO. If the reviewer's reference is to some how differential impact of rIFN β on WT vs. IFNBKO, this is a repeated and constant observation, the effect of rIFN β on IFNBKO neurons (cells in general) is lower as expected since they lack IFNB gene, which is normally induced as a result of IFN β /IFNAR signaling to trigger further downstream genes!

- The conclusion - enhanced FoxA1 nuclear translocation - only from the Fig. 4f is an overstatement. Western blotting with nuclei followed by quantitative evaluation is suggested.

We appreciated this suggestion and the experiment is performed and the results are included in current figure 4f which solidifies the conclusion.

- Fig. 4f and other IF panels with inset letters. The letters are too small. These are addressed to the best of our possibility to adhere also to the page size and multi-panel figures.
- Fig. 4m. An empty vector control is required. Expression levels of PD-L1 have to be shown too.

The control empty vector is included and it is shown in current figure 4p.

- How the peaks were gated in Fig. 4h, k; 5g to consider some cells to be positive?

Fig 4h and 4k and 5g are the histograms showing the comparison of antibody positive shift (anti-FoxA1 or anti-PDL1) to isotype control (iso), that is how the gate/ positive shift is determined i.e. shift from isotype control is considered as positive.

- Were the results in Fig. 4, j, l, m and Fig. 5h obtained from the histograms in Fig. 4h, k; 5g?

Figure 4j (current fig 4k) is showing the FACS result of FoxA1+Treg cells generation after KD of FoxA1 on CGNs.

Figure 4h (current fig 4i) is FACS histograms showing expression levels of FoxA1 and PD-L1 after KD of FoxA1 vs. control KD (siCtrl) in neurons.

Figure 4l (current fig 4m) is showing the FACS result of FoxA1+Treg cells generation after KD of PD-L1 vs. control KD (siCtrl) of CGNs, data is not obtained from 4k. and histogram 4k (current fig 4l) is indicating the expression of PD-L1 after KD of PD-L1 vs. control (UNC-1 siRNA) on neurons.

Figure 4m (current version is 4o and 4p); current fig. 4o is showing the FACS histogram of successful PDL1 expression vs. control EGFP vector after transfection, and fig 4p is showing the quantifications of FoxA1+Treg cells generation after co-culture of T_{enc} with IfnbKO CGN with control vector or after overexpression of PD-L1 on CGNs.

Figure 5h is quantification of representative FACS histogram shown in figure

5g. All details are explained in the result section.

- Total Akt levels are low in *lfnb*^{-/-} samples in Fig. 5b. Simply not good data. Unfortunately, because of the typo error, we are not sure of the point that reviewer #1 wanted to make. However, these are the actual biological findings in primary neuronal cultures shown in fig 5b, and we think these are very good WB findings, which are indeed consistent in 3 and 3 independent cultures of primary neurons! And confirmed by phosphorylated AKT in the line below that.
- Fig. 5i. IP has to be performed with CGNs. IgG IP control also shows background pAkt. Was this IP clean enough? Need better labeling in the figure; e.g., labeling with Ab names used for IP and WB will help. As suggested, we have performed IP both in N2A neuronal cell line and primary neurons, which are shown in current figure 5i, 5j and 5k.
- Method section. Spell out B27 and PS. Please also write which Ab was used for what, if it is not discussed in the main text, such as NF-200. We appreciate this and now it is addressed.
- Please show primer sequences for qPCR. The primers were purchased from SABiosciences (Qiagen), ready to use as included now in M&M, no sequences are available.
- Five % "milk" was used in a WB blocking solution. What exactly is "milk"? It is addressed.
- Where is the source of 3T3-L1 cells? 3T3-L1 were purchased from ATCC, however, we have changed the figure as suggested and have included the information regarding the use of N2A cells and primary neurons for IP.
- Fig. 2a legend. What exactly are "infiltrating inflammatory cells"? It says CNS, but does it mean spinal cord of the brain? This is addressed both in M&M and results to indicate that all quantifications are performed in spinal cords.
- Labeling "e" is missing in Fig. 2e. It is added.
- Fig. 4j, l, m. How were the data obtained? These are FACS analysis data of FoxA1⁺Treg cells, the gating strategy is shown in figure 3d&e, it is indicated how T cells were gated after co-culture with neurons, FoxA1⁺Treg cells were gated as TCR⁺CD4⁺PD-L1⁺ T cells and FoxA1⁺ cells were enriched in TCR⁺CD4⁺PD-L1⁺ T cells as

shown in R1 gating (3e). Based on this FACS analysis, we obtained percentage of FoxA1+Treg cells after KD of PD-L1, FoxA1 and overexpress of PD-L1 on neurons (figure 4j,l,n-current figures 4k, 4m, 4n and 4p).

- Fig. 5c. What does it mean "ratio." How was it calculated?

Fig 5c is quantification data from 5b, Western blots were quantified with ImageJ software. Ratio means quantification of each individual band divided to (/) quantification of loading control (Vinculin). The information is added to the figure legend.

Reviewer #2 (Remarks to the Author):

In a previous report, I-N et al reported that neurons instruct CD4+ T cells to differentiated to FoxA1+ Treg cells and that this instruction involves production and release of IFN γ . The present report presents experiments to cement and extend the previous observations. It concludes that neurons act directly on invading effector T cells to divert them towards the FoxA1 direction. This is an interesting concept but its formal proof has to be strengthened by additional data. So far, induction may occur indirectly, via glia, especially astrocytes.

We appreciate the positive feedback from the reviewer #2. To address the raised issue regarding the possible involvement of other CNS glial cells, we included data from enriched astrocytes and microglia co-cultured with encephalotogenic T cells. As shown in current figure 3b, in contrast to primary CGNs, both astrocytes and microglial cells were incapable of reverting CD4+T_{enc} cells to generate FoxA1+Tregs.

First, shouldn't NesCre :Ifnar1/fl mice should lack IFNAR on neuron plus astrocytes? Second, the protocols used to generate neuronal cultures permits growth of glia cells, mostly astrocytes, thus an estimate of this contribution should be given. A GFAP staining of co-cultures should be shown at least. The reviewer #2 is correctly referring to the possibility of at least a population of astrocytes also being IFNAR KO, indeed this was the reason that we addressed the question by establishing purified cultures which did not result in FoxA1+Treg generation when pure IFN γ -competent astrocytes were co-cultured with T_{enc} cells. As suggested the images of neuronal cultures is included in the current figure 3a stained with beta III tubulin for neurons and GFAP for astrocytes, which is also quantified in 3 independent cultures and the result of quantification are included (98.3 \pm 0.28% purity of neurons are achieved).

There is also a query related to the site of conversion in the CNS. Apparently, FoxA1+ T cells are differentiating and residing near neuronal bodies, placed in the grey matter, with very sparse infiltration. The perivascular areas of the white

matter harboring the mass of inflammatory cells, are remote.

This is an interesting point, but currently there is no evidence for strict requirements for cell body and not axonal (even dendrite) contact to determine conversion. In fact, as also shown in fig 1 and 2, in addition to mass inflammatory cells observed in the white matter along neuronal fibers, there are accountable parenchymal migrating T cells. And relevant to this question, indeed FoxA1+Treg cells are detectable in the white matter (fig 1b, 2c and 2d) but also in grey matter (current fig. 1e). Of note as shown in cerebellum in figure 1i, PDL1 is expressed both on neuronal soma (clear in Purkinje cells) and also axons. In culture neurons are also positive for PDL1 both in soma and all along axons (Liu et al. 2013, J. Neuroscience).

Another important issue: previous work by others pointed to the importance of neuronal electric activity. Would silencing of neurons (e.g. by tetrodotoxin) cancel induction?

In agreement with the interesting raised point, we have also shown earlier that neurons interacting with encephalitogenic T cells are electrically active (Liu et al 2006, Nature Medicine). We performed experiments addressing this question, and indeed the results showed that neurons are required to be electrically active to exert their regulatory capacity and generate FoxA1+Treg cells. These results are now included in current figure 3c.

Most T cells invade the spinal cord and brain areas remote of neuronal bodies; how could they induce FoxA1 in T cells?

Please see discussion of this point above under the point 3 raised by the reviewer #2

Less fundamental:

Fig.1 should be described in more detail: a) which brain region shown?

How was infiltration quantified? Also give EAE grades. Line 75: “did NOT find apparent FoxA1+ Tregs” contradicts to Fig.1c with 20 vs 35% of such cells, and line 75. Where is “neuronal FoxA1” shown (Fig.1d is not helpful)? Neuronal bodies sit in spinal cord grey matter, which is not a predilection of T cells.

All these issues are now better addressed in M&M, results and figure legends.

The current figure legend for fig 1b is clarifying that the IF images are from spinal cords which are quantified in fig 1c and 1d. The EAE scores are shown in fig 1a and specifications are discussed in M&M and results. The formulations are also revised. Figure 1d is specifically addressing your point regarding the interaction between neuronal soma in the grey matter and infiltrating T cells.

Legend to Fig.4: (f) and (g) are interchanged. NF200 expression mostly in nucleus, why?

The figures and legend are reorganized.

We agree that this is an interesting observation, which we did not find any

explanation in the literature, we currently do not know the role of NF200 in nucleus, however neurofilaments are used as ducking /carriage to transport many signaling molecules including autophagy related proteins, it is likely that NF200 is involved in the similar functions.

Forced expression of PDL-1 overrides IFN β deficiency. Why would PDL1 on microglia and other cells (T cells included) fail to substitute? Is there an additional neuronal factor at work?

Additional neuronal factor/s is plausible (especially upon PDL1 regulation), we have tried to discuss this in the discussion section. We favor the viewpoint that some APCs might act as tolerogenic but not all (as well describe in the literature). In discussion, we speculate that lack of MHC II molecules on neurons (lack of signal 1 for T cell activation), but existence of costimulatory signal (signal 2) together with physiological levels of IFN β leads to induction of FoxA1 and its target gene PDL1 which is in turn sufficient to trigger a tolerogenic signal in T cells, i.e. in contrast to an activation signal. While astrocytes and microglial cells under inflammatory conditions in vivo (and in vitro dissection and culture which could provide similar conditions) upregulates expression of MHC II molecules and all co-stimulatory molecules (Teige et al 2006 JI) require for conducting an activation signal in T cells rather than tolerogenic. Of note we do not exclude a possible mapping and defining a subpopulation of glial cells which under normal conditions could exert similar properties.

** See Nature Research's author and referees' website at www.nature.com/authors for information about policies, services and author benefits

This email has been sent through the NPG Manuscript Tracking System NY-610A-NPG&MTS

Confidentiality Statement:

This e-mail is confidential and subject to copyright. Any unauthorised use or disclosure of its contents is prohibited. If you have received this email in error please notify our Manuscript Tracking System Helpdesk team at <http://platformsupport.nature.com> .

Details of the confidentiality and pre-publicity policy may be found here <http://www.nature.com/authors/policies/confidentiality.html>

Privacy Policy | Update Profile

REVIEWERS' COMMENTS:

Reviewer #1 (Remarks to the Author):

All the inquiries have been answered. The manuscript has been greatly improved.

The authors answered questions that both reviewers asked. Although some experiments were not performed as asked by reviewers, new data was added and the manuscript was strengthened. For better clarification, some questions need to be discussed or mentioned in the manuscript. These includes: (1) how T cells could induce FoxA1, although they invade the CNS areas remote of neuronal bodies, (2) reason to use B10RIII mice, (3) why IFN-alpha was not touched upon in this study, and (4) NF200 in nucleus.

We thank the reviewers for the final suggestions. Please below find how each point has been addressed in the manuscript (highlighted in blue).

REVIEWERS' COMMENTS:

Reviewer #1 (Remarks to the Author):

All the inquiries have been answered. The manuscript has been greatly improved.

The authors answered questions that both reviewers asked. Although some experiments were not performed as asked by reviewers, new data was added and the manuscript was strengthened. For better clarification, some questions need to be discussed or mentioned in the manuscript. These includes:

(1) how T cells could induce FoxA1, although they invade the CNS areas remote of neuronal bodies,

This is addressed in the result section by addition of the following to the fig. 1b “Of note, FoxA1⁺T_{regs} were often detected not only in perivascular space (Fig. 1b) but also in parenchyma adjacent to neuronal soma (Fig. 1e), this provides neurons with possibility to interact with T cells and induce their FoxA1 expression both via molecules expressed on neuronal cell bodies or axons in the areas that T cells invade the CNS remote from neuronal bodies as well as closed to the neuronal bodies upon migration in the parenchyma.”

(2) reason to use B10RIII mice,

We have addressed this issue as suggested in the Methods part under the subheading of “*Induction of neuroinflammation; Experimental autoimmune encephalomyelitis (EAE)*”. The following text is added: “The reason for utilizing two different models of EAE either in C57BL/6 or C57BL/10RIII backgrounds was to address both chronic EAE induced by MOG₃₅₋₅₅, which is commonly used because of the availability of many transgenic strains in the C57BL/6 genetic background, and to utilize an EAE model induced by MBP₈₉₋₁₀₁, which is the model for relapsing-remitting MS in C57BL/10RIII background^{10,13}. Additionally, such an approach also excludes the possible biased introduced by slight differences in the genetic backgrounds (C57BL/6 versus C57BL/10RIII) rather than the function of the target genes, here *Ifnb* and *Ifnar*.”

(3) why IFN-alpha was not touched upon in this study, and

We have addressed this issue in the result section as following:

“IFN β share many functional similarities with IFN α as they share the same receptor, IFNAR, however they also differ in many of their functions including their different efficiencies as treatments for several diseases. Although it is not well described how IFN β might regulate IFN α , it is previously reported that IFN β is required for production of IFN α in fibroblast¹³, and we have not detected any compensatory mechanisms in neurons when only IFN β is deleted⁹. Although IFN α might have additional or differential effects independent of IFN β , this has not been observed related to the neuronal activity. Moreover since there are several alleles for *Ifna* resulting in production of many active IFN α isoforms, this is imposing much more complexity to investigate the plausible role for IFN α isoforms in relation to the neuronal functions and hence it has not been the focus of current study.”

(4) NF200 in nucleus.

This is addressed in the results part under fig.4g by adding the following explanation:

“Of note, the neuronal marker NF200 was also detected in the nucleus, the precise function of which is not well defined, an explanation for this observation might be the fact that neurofilaments are used as ducking or carriage proteins to transport signaling molecules.”